# Fortuity in the D1-D5 system

Chi-Ming Chang[a,b], Ying-Hsuan Lin[c], and Haoyu Zhang[d]

[a] *Yau Mathematical Sciences Center (YMSC), Tsinghua University, Beijing 100084, China*

[b] *Beijing Institute of Mathematical Sciences and Applications (BIMSA)*
*Beijing 101408, China*

[c] *Jefferson Physical Laboratory, Harvard University, Cambridge, MA 02138, USA*

[d] *George P. & Cynthia Woods Mitchell Institute for Fundamental Physics and Astronomy,*
*Texas A&M University, College Station, TX 77843, USA*

cmchang@tsinghua.edu.cn, yhlin@alum.mit.edu, zhanghaoyu@tamu.edu

## Abstract

We reformulate the lifting problem in the D1-D5 CFT as a supercharge cohomology problem, and enumerate BPS states according to the fortuitous/monotone classification. Focusing on the deformed $T^4$ symmetric orbifold theory, cohomology classes in the $N = 2$ theory are explicitly constructed and matched with the exact BPS partition function. For general $N$, an infinite set of monotone cohomology classes are characterized and conjectured to be exhaustive. We further describe how to assemble BPS states at smaller $N$ into BPS states at larger $N$, and interpret their holographic duals as black hole bound states and massive stringy excitations on smooth horizonless (e.g. Lunin-Mathur) geometries.

# 1  Introduction and summary of results

Recent progress in identifying the non-graviton states [1–6], also dubbed fortuitous states [7], in the $\frac{1}{16}$-BPS sector of $\mathcal{N} = 4$ supersymmetric Yang-Mills (SYM) has significantly enhanced our understanding of black hole microstates in the framework of the AdS/CFT correspondence [8]. This development goes beyond the standard index counting [9–11] and examines the supercharge $Q$-cohomology [9, 12, 13], which is isomorphic to the space of BPS states, and offers much refined information about the microstates. In a nutshell, fortuitous states have a distinctive property that prevents them from remaining BPS at large $N$, and yet they dominate the entropy. Based on these attributes, it was argued and conjectured in [7] that fortuitous states are dual to typical black hole microstates, while graviton states are dual to perturbative excitations and smooth horizonless geometries.

In $\mathcal{N} = 4$ SYM, fortuitous states arise due to non-trivial trace relations at finite $N$ among gauge-invariant operators. The complexity of these trace relations makes it challenging to analytically understand and systematically classify these states. To gain deeper insights into this phenomenon, it is natural to explore other well-studied holographic models. This paper focuses on the D1-D5 CFT, which is central to $\text{AdS}_3 \times \text{S}^3$ holography and offers a more accessible framework for investigating finite $N$ effects and the associated fortuitous states. This is also the cradle of the fuzzball program (see [14] for a review), to which the fortuity conjecture of [7] poses a major challenge. We believe that the framework laid out in this paper will shed light on the microscopic nature of the numerous horizonless geometries constructed in [15–26].

String theory on $\text{AdS}_3 \times \text{S}^3 \times T^4$ is dual to the symmetric orbifold of the $T^4$ sigma model with exactly marginary deformations [8, 27–31]. At the orbifold point, the degeneracy of low-lying states grows exponentially with energy, in contrast to the sub-exponential growth expected in supergravity [32–34]. However, it is believed that exactly marginal deformations give most of these low-lying state large anomalous dimensions, rendering the spectrum compatible with the supergravity interpretation [35–37].

These anomalous dimensions can be studied using conformal perturbation theory. The leading nontrivial contribution to the lifting matrix arises at second order and can be computed using the method introduced by Gava and Narain [38–40]. This approach leverages the supersymmetry algebra, expressing the lifting matrix $\Delta$ as an anti-commutator

$$\{Q, Q^\dagger\} = \Delta \,. \tag{1.1}$$

The computation then reduces to evaluating the matrix elements of the supercharge $Q$ at *first* order in conformal perturbation theory, which corresponds to a three-point function calculation at the orbifold point. Other studies have directly examined the lifting matrix at

second order in conformal perturbation theory [41–44], which involves calculating four-point functions at the orbifold point. The machinery for computing correlations in the symmetric orbifold theory was laid out in [45, 46] based on the covering space method of [47]. Most of these analyses focus on states with relatively small quantum numbers, since extending these results to states with higher quantum numbers is challenging due to the increased complexity introduced by the higher fractional modes. Similarly, extending these studies to higher-order perturbation theory is difficult as it necessitates the evaluation of higher-point functions.

A review of the D1-D5 CFT and the lifting of BPS states can be found in Section 2.

For BPS states (states with $\Delta = 0$), the supercharge $Q$ cohomology offers a simpler framework for studying their lifting, since by a standard argument from Hodge theory, they are in one-to-one correspondence with $Q$-cohomology classes. In $\mathcal{N} = 4$ SYM [9,12,13], it was argued based on the rigidity of $Q$-cohomology that the BPS states at one loop remain BPS to all loops, perhaps even non-perturabtively. While we do not have a rigorous argument, the relative rigidity of $Q$-cohomology encourages us to propose the following conjecture:

**Conjecture 1** (Non-renormalization). *The spectrum of BPS states in the D1-D5 CFT is determined exactly by the lifting matrix at second order, without any higher-order corrections.*

A similar phenomenon was observed in the K3 CFT [48].

To gain more evidence for this conjecture, in Section 3, we systematically study the (finite $N$) supercharge $Q$-cohomology under first-order conformal perturbation theory. After laying out the general machinery in Sections 3.1 and 3.2, we apply it specifically to $N = 2$ in Section 3.3, and our results for small charges precisely match the known BPS partition function, which was computed using a bootstrap method that assumed no symmetry enhancement under generic deformations [49, 41]. An analysis of the large $N$ scaling can be found in Section 3.4. In the infinite $N$ limit (with charges fixed), the supercharge $Q$-cohomology classes are expected to be dual to non-interacting (multi-)gravitons in vacuum $\mathrm{AdS}_3 \times \mathrm{S}^3 \times T^4$. The $Q$-operator in this limit was recently studied in [50], and found to be nicer than for finite $N$, e.g. $Q_{N=\infty}$ obeys the Leibniz rule when acting on multi-cycle states. We conjecture that taking large $N$ and taking cohomology commutes, i.e. the $Q_{N=\infty}$-cohomology is isomorphic to the $N \to \infty$ limit of the $Q$-cohomology. This sets the stage for our investigation in Section 4.2.

At finite $N$, a significant simplification of the supercharge $Q$-cohomology in the D1-D5 CFT compared to $\mathcal{N} = 4$ SYM is that the trace relations in $\mathcal{N} = 4$ SYM are replaced by much simpler relations: the stringy exclusion principle [27]. In the symmetric orbifold theory, the twisted sectors are labeled by cycle shapes, and the stringy exclusion principle eliminates cycle shapes where the total length of nontrivial cycles exceeds $N$. Following [7], the supercharge $Q$-cohomology classes are classified into two categories: the monotone

cohomology classes, obtained by imposing the stringy exclusion principle on the infinite $N$ cohomology classes, and the fortuitous cohomology classes, mapped by $Q$ to states with cycle shapes whose total length of nontrivial cycles exceeds $N$ (see Section 4.1). A series of conjectures regarding the bulk duals of the monotone and fortuitous cohomology classes was proposed in [7]. In the D1-D5 context, the key statements include the following:

- The monotone cohomology classes correspond to bulk states arising from the quantization of smooth horizonless geometries [51–54], specifically the Lunin-Mathur geometries [15–18] and the superstrata geometries [19–26].

- The fortuitous cohomology classes correspond to typical microstates of the 3-charge (D1-D5-P) black holes [55, 56].

The counting of monotone cohomology classes can be studied by imposing the stringy exclusion principle on the $Q_{N=\infty}$-cohomology, resulting in the monotone partition function, which matches exactly with the counting of bulk states corresponding to superstrata geometries [53, 54], as shown in Section 4.2. Fortuitous cohomology classes, on the other hand, are much harder to study. In Sections 4.3 and 4.4, we focus on $N = 2$ and perform a comprehensive search for fortuitous states in small charge sectors. We provide concrete data demonstrating that the number of fortuitous cohomology classes dominates over the monotone ones, as illustrated in Figure 1, and present explicit representatives for the leading fortuitous cohomology classes.

In Section 5.1, we construct two-cycle composite BPS states by taking products of two single-cycle BPS states (of cycle length $w_1$ and $w_2$, respectively) and performing certain projections. Depending on whether each constituent is monotone or fortuitous, such states have different gravitational interpretations, and are the subject of Section 5.2. When at least one constituent is fortuitous, the composite is also fortuitous in the $N = w_1 + w_2$ theory. For two fortuitous constituents, the composites naturally have the interpretation of black hole bound states or near horizon geometries of two-centered black hole solutions [57–59]. Instead, if one is fortuitous and the other monotone, then in the limit $w_2 \gg w_1$, the composite can be interpreted as a fortuitous excitation probing a monotone background. A length-$w_1$ fortuitous state in the large $N$ limit is dual to a massive stringy excitation in the vacuum $\text{AdS}_3 \times \text{S}^3 \times T^4$. Thus, it is natural to propose that the composite state, in the large $N$ limit ($N = w_1 + w_2 \gg w_1$), is dual to a BPS stringy excitation on a smooth horizonless geometry corresponding to the cycle-length $w_2$ monotone state. It is surprising that the non-BPS stringy excitations in $\text{AdS}_3 \times \text{S}^3 \times T^4$ vacuum can become BPS in smooth horizonless geometries, such as the superstrata geometries. Although the composite fortuitous states we construct should only constitute a minority of all fortuitous states, their holographic

interpretation provides hints on how general fortuitous states might be understood in the bulk string theory.

# 2 Review of the D1-D5 CFT

The D1-D5 CFT describes the low energy limit of a supersymmetric configuration of D1- and D5-branes in type IIB string theory on $\mathbb{R}^6 \times \mathcal{M}$, where $\mathcal{M}$ can be $T^4$ or K3. Its conformal moduli correspond to the geometric and flux deformations.

## 2.1 Symmetric orbifold of the $T^4$ CFT

The $T^4$ CFT is a system of four free bosons, four free left-moving fermions, and four free right-moving fermions,

$$X^\mu(z, \bar{z}), \quad \psi^\pm(z), \quad \bar{\psi}^\pm(z), \quad \tilde{\psi}^\pm(\bar{z}), \quad \tilde{\bar{\psi}}^\pm(\bar{z}), \tag{2.1}$$

where $\mu = 1, \cdots, 4$. We will mainly consider the theory in the Neveu–Schwarz (NS) sector, where the fermions carry half-integer modes. The free bosons $X^\mu$ are valued in $T^4$, i.e. satisfy the identification $X^\mu \sim X^\mu + 2\pi R^\mu$, where $R^\mu$ are the radii of $T^4$. Let us denote the Hilbert space in the NS sector by $\mathcal{H}$, which is simply the Fock space generated by the free boson and fermion creation operators acting on the NS sector ground state $|1\rangle$.

Now, consider the symmetric orbifold theory $\mathrm{Sym}^N(T^4)$, which contains an untwisted sector and several twisted sectors. The Hilbert space of the untwisted sector is a (graded) symmetric tensor product of $\mathcal{H}$,

$$S^N \mathcal{H} = \Big( \underbrace{\mathcal{H} \otimes \mathcal{H} \otimes \cdots \otimes \mathcal{H}}_{N \text{ times}} \Big)^{S_N}, \tag{2.2}$$

and each twisted sector is labeled by a cycle-shape $p$ (an integer partition of $N$),

$$p = (\underbrace{1, 1, \cdots, 1}_{N_1 \text{ times}}, \underbrace{2, 2, \cdots, 2}_{N_2 \text{ times}}, \cdots) = (1^{N_1}, 2^{N_2}, \cdots), \quad N = N_1 + 2N_2 + 3N_3 + \cdots. \tag{2.3}$$

The untwisted sector could be regarded as the twisted sector with $N_1 = N$. The Hilbert space $\mathcal{H}_p$ can be explicitly constructed as follows. Given a group element $g \in S_N$ belonging to the conjugacy class specified by the cycle shape $p$, let $\mathcal{H}_g$ denote the Hilbert space obtained

by quantizing the tensor product theory $(T^4)^N$ on $S^1$ with a twisted boundary condition

$$X^{(i)}(\sigma + 2\pi) = X^{(g(i))}(\sigma), \quad \psi^{(i)}(\sigma + 2\pi) = \psi^{(g(i))}(\sigma), \tag{2.4}$$

where $X$ and $\psi$ collectively denote the free fields in (2.1), and $\sigma \in [0, 2\pi)$ is the coordinate of the $S^1$. Given $|\Psi\rangle \in \mathcal{H}_g$, we obtain a state $|\{\Psi\}\rangle \in \mathcal{H}_p$ by summing over the $S_N$ orbit:

$$|\{\Psi\}\rangle = \sum_{\Psi' \in S_N(\Psi)} |\Psi'\rangle = \frac{1}{|G_\Psi|} \sum_{g \in S_N} g \, |\Psi\rangle \in \mathcal{H}_N, \tag{2.5}$$

where $S_N(\Psi)$ is the $S_N$ orbit of the state $|\Psi\rangle$, and $|G_\Psi|$ is the cardinality of the invariant subgroup $G_\Psi \in S_N$ preserving the state $|\Psi\rangle$. For studying the supercharge cohomology, it is convenient to choose the normalization

$$\langle \Psi | \Psi \rangle = 1, \tag{2.6}$$

and hence the state $|\{\Psi\}\rangle$ is not normalized.

The full Hilbert space is a direct sum over the Hilbert spaces of all twisted sectors,

$$\mathcal{H}_N = \bigoplus_p \mathcal{H}_p. \tag{2.7}$$

There is an isomorphism

$$\iota: \quad \mathcal{H}_{p=(1^{N_1}, 2^{N_2}, \cdots)} \cong \bigotimes_{w=1}^{\infty} S^{N_w} \mathcal{H}_{(w)}^{\mathbb{Z}_w}, \tag{2.8}$$

where $\mathcal{H}_{(w)}$ is the Hilbert space of the single-copy $T^4$ CFT on a circle with $w$ times the radius (in particular, $\mathcal{H}_{(1)} = \mathcal{H}$), and $\mathcal{H}_{(w)}^{\mathbb{Z}_w}$ is the $\mathbb{Z}_w$-invariant sector in $\mathcal{H}_{(w)}$, i.e. the sector with integer or half-integer spin $\ell = h - \tilde{h}$ for bosonic or fermionic states. More explicitly, the Hilbert space $\mathcal{H}_{(w)}$ is a Fock space generated by the bosonic and fermionic creation operators with fractional modes,

$$\alpha_{n+\frac{m}{w}} = \oint_{z=0} \partial X^{[m]}(z) \, z^{n+\frac{m}{w}} \, dz, \quad \psi_{r+\frac{m}{w}} = \oint_{z=0} \psi^{[m]}(z) \, z^{r+\frac{m}{w}-\frac{1}{2}} \, dz,$$

$$X^{[m]}(z) = \frac{1}{\sqrt{w}} \sum_{j=1}^{w} e^{-2\pi i \frac{m}{w} j} X^{(j)}(z), \quad \psi^{[m]}(z) = \frac{1}{\sqrt{w}} \sum_{j=1}^{w} e^{-2\pi i \frac{m}{w} j} \psi^{(j)}(z), \tag{2.9}$$

where $X^{(j)}$ and $\psi^{(j)}$ satisfy the boundary condition $X^{(j)}(e^{2\pi i} z)\sigma_w(0) = X^{(j+1)}(z)\sigma_w(0)$, and similarly for $\psi$ and $j = 1, \cdots, w$ with identification $w + 1 \sim 1$. Under the state/operator

correspondence, the ground state in $\mathcal{H}_{(w)}$ is dual to a $\mathbb{Z}_w$ twisted operator $\sigma_w$, and $\mathcal{H}_{(w)}$ can be equivalently regarded as the radially quantized Hilbert space with $\sigma_w$ inserted at the origin.

The inverse map $\iota^{-1}$ of the Hilbert space isomorphism (2.8) is given by summing over the $S_N$ orbit. By definition, orbifold/gauging restricts to the $S_N$ invariant subspace, which is isomorphic to the $S_N$ orbit,[1] and hence the inverse map is surjective. It is also injective by noting that the $S_N$ orbit of an arbitrary state on the right consists of mutually orthogonal elements.

## 2.2 Contracted large $\mathcal{N} = 4$ and BPS Clifford modules

The $T^4$ CFT has the contracted large $\mathcal{N} = 4$ superconformal symmetry [60] with central charge $c = 6$, generated by the left-moving currents

$$T, \quad G^\pm, \quad G'^\pm, \quad K^\pm, \quad K^3, \tag{2.10}$$

and

$$\alpha^i = \partial(X^{2i-1} + iX^{2i}), \quad \bar{\alpha}^i = \partial(X^{2i-1} - iX^{2i}), \quad \psi^\pm, \quad \bar{\psi}^\pm, \tag{2.11}$$

where $i = 1, 2$. Here, $T(z)$ is the stress tensor, $G^\pm(z), G'^\pm(z)$ are the four superconformal currents, $K^\pm(z), K^3(z)$ are the spin-1 currents that generate the $\mathfrak{su}(2)$ Kac-Moody algebra at level 1, $\alpha^i(z), \bar{\alpha}^i(z)$ are the spin-1 currents that generate the $\mathfrak{u}(1)^4$ Kac-Moody algebra, and finally $\psi^\pm(z), \bar{\psi}^\pm(z)$ are the spin-$\frac{1}{2}$ free fermions. The currents in (2.10) are composites of those in (2.11) with the explicit formulae given in (A.2). Setting $\alpha^i$, $\psi^\pm$, and $\bar{\psi}^\pm$ to zero is a consistent truncation that recovers the small $\mathcal{N} = 4$ algebra. The symmetric orbifold theory inherits the superconformal symmetry with the generators given by the sum of the generators over all copies; for example,

$$T = \sum_{i=1}^{N} T^{(i)}, \tag{2.12}$$

and similarly for the other currents.

The *superconformal primary* states (in the NS sector) are defined as the states annihilated by all the positive modes of the currents (2.10), (2.11) and their right-moving counterparts. Let us focus on the right-moving part. The (right-moving) *chiral primary* states are the

---

[1]This isomorphism is a general fact about $G$-spaces when $G$ is finite.

superconformal primaries further annihilated by the supercharges

$$Q := \widetilde{G}^+_{-\frac{1}{2}}, \quad Q' := \widetilde{G}'^+_{-\frac{1}{2}}. \tag{2.13}$$

Together with their Hermitian conjugates $Q^\dagger = \widetilde{G}^-_{\frac{1}{2}}$ and $Q'^\dagger = \widetilde{G}'^-_{\frac{1}{2}}$, they satisfy the supersymmetry algebra

$$\{Q, Q^\dagger\} = \{Q', Q'^\dagger\} = 2(\widetilde{L}_0 - \widetilde{K}^3_0) =: \Delta, \tag{2.14}$$

and the BPS condition

$$\tilde{h} = \tilde{j}, \tag{2.15}$$

where $\tilde{j}$ is the eigenvalue of $\widetilde{K}^3_0$.

Since the $\mathfrak{u}(1)^4$ charges do not appear in the BPS formula (2.15), for a generic $T^4$, the states with nonzero momentum or winding necessarily violate the BPS condition; in other words, momentum and winding increase $\tilde{h}$ but leave $\tilde{j}$ intact.[2] As we are interested in BPS states, we can and will henceforth restrict ourselves to the sector with zero momentum and winding.

In the contracted large $\mathcal{N} = 4$ superconformal algebra, besides the supercharges $Q$ and $Q'$, there are four spin-$\frac{1}{2}$ fermionic generators that satisfy the BPS condition (2.15) and generate a Clifford algebra[3]

$$\{\tilde{\psi}^+_{-\frac{1}{2}}, \tilde{\psi}^-_{\frac{1}{2}}\} = -1, \quad \{\tilde{\tilde{\psi}}^+_{-\frac{1}{2}}, \tilde{\tilde{\psi}}^-_{\frac{1}{2}}\} = 1. \tag{2.16}$$

In the zero momentum and winding sector, the supercharges $Q$ and $Q'$ commute with this Clifford algebra,

$$\begin{aligned}
\{Q, \tilde{\psi}^+_{-\frac{1}{2}}\} &= \{Q, \tilde{\tilde{\psi}}^+_{-\frac{1}{2}}\} = \{Q', \tilde{\psi}^+_{-\frac{1}{2}}\} = \{Q', \tilde{\tilde{\psi}}^+_{-\frac{1}{2}}\} = 0, \\
\{Q, \tilde{\psi}^-_{\frac{1}{2}}\} &= -\tilde{\alpha}^2_0, \quad \{Q, \tilde{\tilde{\psi}}^-_{\frac{1}{2}}\} = \tilde{\alpha}^2_0, \quad \{Q', \tilde{\psi}^-_{\frac{1}{2}}\} = \tilde{\alpha}^1_0, \quad \{Q', \tilde{\tilde{\psi}}^-_{\frac{1}{2}}\} = -\tilde{\alpha}^1_0,
\end{aligned} \tag{2.17}$$

and analogously for $Q^\dagger$ and $Q'^\dagger$, which follows by taking the Hermitian conjugate of (2.17). Therefore, the $\tilde{\psi}^+_{-\frac{1}{2}}$ and $\tilde{\tilde{\psi}}^+_{-\frac{1}{2}}$ descendants of a chiral primary state also satisfy the BPS condition, forming a four-dimensional (quartet) representation of the Clifford algebra (2.16).

---

[2]For special $T^4$ moduli, there can be states with nonzero momentum and winding, in such a way that $h$ is increased but not $\tilde{h}$, so that the states remain BPS on the right.

[3]The bosonic generators $\widetilde{K}^\pm_{\mp 1}$ also satisfy the BPS condition. The chiral primary states are the lowest weight states of a $(\frac{c}{6} - 2\tilde{j} + 1)$-dimensional representation of the SU(2) generated by $K^+_{-1}$, $K^-_1$ and $K^3_0 - \frac{c}{12}$. In the $T^4$ symmetric orbifold theory, we have $\widetilde{K}^{+(i)}_{-1} = \tilde{\psi}^{+(i)}_{-\frac{1}{2}} \tilde{\tilde{\psi}}^{+(i)}_{-\frac{1}{2}}$, when acting on right-moving chiral primaries. Consequently, $K^{+(i)}_{-1}$, when applied to BPS states, does not generate new BPS states.

If we let $|w_{--}\rangle$ denote the bottom component, which is annihilated by $\tilde{\psi}^-_{\frac{1}{2}}$ and $\tilde{\bar{\psi}}^-_{\frac{1}{2}}$, then the explicit R-charges are

$$\text{BPS Clifford quartet :} \quad \begin{bmatrix} |w_{++}\rangle \\[4pt] |w_{+-}\rangle \\[4pt] |w_{-+}\rangle \\[4pt] |w_{--}\rangle \end{bmatrix}, \quad \tilde{j} = \begin{bmatrix} \frac{w+1}{2} \\[4pt] \frac{w}{2} \\[4pt] \frac{w}{2} \\[4pt] \frac{w-1}{2} \end{bmatrix}. \tag{2.18}$$

In other words, $|w_{\alpha\beta}\rangle$ has $\tilde{j} = \frac{2w+\alpha+\beta}{4}$.

The *chiral-chiral primary* states are the superconformal primaries further annihilated by both (2.13) and its left-moving counterpart. The only chiral-chiral primary state in the $T^4$ CFT is the vacuum state with dimension $h = \tilde{h} = 0$. The chiral-chiral primary state in the single-cycle twisted sector $\mathcal{H}^{\mathbb{Z}_w}_{(w)}$ is the vacuum state of the single-copy theory on a circle with $w$ times the radius, with weights and (R-symmetry) spins

$$h = \tilde{h} = \frac{w-1}{2} = j = \tilde{j}. \tag{2.19}$$

The chiral-chiral primary states in the multi-cycle twisted sectors are exhaustively given by the products of the chiral-chiral primary states in the single-cycle twisted sectors; in particular, they are all Lorentz scalars. The chiral-chiral primary states and their $\psi^+_{-\frac{1}{2}}$, $\bar{\psi}^+_{-\frac{1}{2}}$, $\tilde{\psi}^+_{-\frac{1}{2}}$, $\tilde{\bar{\psi}}^+_{-\frac{1}{2}}$ descendants give the full set of $\frac{1}{2}$-BPS states satisfying the BPS condition (2.15) and the left-moving BPS condition

$$h = j. \tag{2.20}$$

Let us denote the $\frac{1}{2}$-BPS states in $\mathcal{H}^{\mathbb{Z}_w}_{(w)}$ by

$$|w_{\pm\pm,\pm\pm}\rangle, \tag{2.21}$$

where the two pairs of $\pm$'s separately denote the left-moving and right-moving quartets with $|w_{--,--}\rangle$ being the chiral-chiral primary state. A general multi-cycle $\frac{1}{2}$-BPS state is given by products of the single-cycle $\frac{1}{2}$-BPS states.

The $\frac{1}{4}$-BPS states satisfying the right-moving BPS condition (2.15) can be constructed by acting the left-moving creation operators acting on the $\frac{1}{2}$-BPS states. Let us choose a basis for this $\frac{1}{4}$-BPS space as

$$|w_{i,\alpha\beta}\rangle, \tag{2.22}$$

where $\alpha, \beta = \pm$ and the index $i$ labels the different combinations of left-moving generators

acting on $|w_{--,\pm\pm}\rangle$. Let us work with the following basis of multi-cycle $\frac{1}{4}$-BPS states:

$$\left|\{(w_1)_{i_1,\alpha_1\beta_1}^{N_1}, \cdots, (w_\nu)_{i_\nu,\alpha_\nu\beta_\nu}^{N_\nu}\}\right\rangle, \tag{2.23}$$

where all $(w_n)_{i_n,\alpha_n\beta_n}$ are distinct and only nontrivial cycles are included in the list. The total length of non-trivial cycles is $\ell := \sum_n N_n w_n \leq N$. The state (2.23) is again given by summing over the $S_N$ orbit as in (2.5), where the number of terms in the sum is

$$\mathcal{N} := \frac{N!}{(N-\ell)!} \times \prod_{n=1}^{\nu} \frac{1}{N_n!\,w^{N_n}}, \tag{2.24}$$

and $N_n$ is the multiplicity of $w_{i,\alpha\beta}$ appearing in (2.23) [61]. We let the states before the sum be normalized, and hence the norm of the state (2.23) is

$$\left|\left|\{(w_1)_{i_1,\alpha_1\beta_1}, \cdots, (w_n)_{i_\nu,\alpha_\nu\beta_\nu}\}\right\rangle\right|^2 = \mathcal{N}. \tag{2.25}$$

Throughout the remainder of this paper, we will omit trivial cycles when expressing the state in the form (2.23). We denote the space spanned by the states (2.23) as

$$V_{(w_1,\cdots,w_\nu)} = \mathrm{span}\left(\left|\{(w_1)_{i_1,\alpha_1\beta_1}^{N_1}, \cdots, (w_\nu)_{i_\nu,\alpha_\nu\beta_\nu}^{N_\nu}\}\right\rangle\right) \tag{2.26}$$

and refer to it as the *free* BPS sector—the BPS sector in the free orbifold theory.[4]

The right-moving R-charge of the state (2.23) is the sum of individual single-cycle R-charges (2.18),

$$\tilde{j} = \sum_{n=1}^{\nu} N_n \times \frac{2w_n + \alpha_n + \beta_n}{4}, \tag{2.27}$$

satisfying

$$\sum_{n=1}^{\nu} N_n \times \frac{w_n - 1}{2} \leq \tilde{j} \leq \sum_{n=1}^{\nu} \times \frac{w_n + 1}{2}, \tag{2.28}$$

or

$$\frac{\ell - \nu}{2} \leq \tilde{j} \leq \frac{\ell + \nu}{2}, \tag{2.29}$$

where $\ell$ and $\nu$ are the total length and the total number of the non-trivial cycles.

---

[4]The free BPS sector here is analogous to the classically-BPS sector in the $\mathcal{N} = 4$ SYM [1].

## 2.3 Lifting under conformal perturbation theory

Consider deforming the symmetric product theory by an exactly marginal operator $\Phi(z, \bar{z})$,

$$S = S_{Sym^N(T^4)} + g \int d^2z \, \Phi(z, \bar{z}). \tag{2.30}$$

We choose $\Phi(z, \bar{z})$ to be the state/operator dual of

$$\frac{i}{\sqrt{2}}(G^-_{-\frac{1}{2}}\widetilde{G}'^-_{-\frac{1}{2}} - G'^-_{-\frac{1}{2}}\widetilde{G}^-_{-\frac{1}{2}}) \, |\{2_{--,--}\}\rangle \,, \tag{2.31}$$

where $|\{2_{--,--}\}\rangle$ contains $N - 2$ trivial cycles and the bottom component of the $\frac{1}{2}$-BPS quartet in the $\mathbb{Z}_2$ twisted sector, with $h = j = \tilde{h} = \tilde{j} = \frac{1}{2}$. In the AdS/CFT correspondence, this particular choice of the exactly marginal operator corresponds to tuning the RR flux in the AdS$_3 \times$ S$^3$ factor [50].

The $\frac{1}{2}$-BPS states are protected under any (contracted large) $\mathcal{N} = (4, 4)$-preserving exactly marginal deformation. To see this, note that their lifting would require the recombination of a pair of chiral-chiral primary states whose spins differ by a half. However, in the $T^4$ symmetric orbifold theory, all chiral-chiral primary states in the theory are scalars, hence such a recombination is not possible. Some $\frac{1}{4}$-BPS states are also protected, while the others acquire anomalous dimensions. A central question in understanding D1-D5 black hole microstates is to describe these unlifted states.

The anomalous dimension is a sum of the left-moving and right-moving anomalous dimensions, $\delta h + \delta\tilde{h}$. The deformation given in (2.30) and (2.31) preserves Lorentz symmetry; hence, the "anomalous spin" $\delta J = \delta\tilde{h} - \delta h$ must be zero. Therefore, it suffices to compute the right-moving anomalous dimension $\delta\tilde{h}$. For a $\frac{1}{4}$-BPS state $|\Psi\rangle$ satisfying (2.15) in the symmetric orbifold theory, we have

$$\delta\tilde{h} \, |\Psi\rangle = (\widetilde{L}_0 - \widetilde{K}_0^3) \, |\Psi\rangle = \frac{1}{2}\{Q^\dagger, Q\} \, |\Psi\rangle = \frac{1}{2}\{Q'^\dagger, Q'\} \, |\Psi\rangle \,, \tag{2.32}$$

where $Q$, $Q'$, and their Hermitian conjugates have vanishing zeroth-order actions on $|\Psi\rangle$. The anomalous dimension $\delta\tilde{h}$ under second-order conformal perturbation can then be computed using the first-order deformed supercharges $Q$ and $Q^\dagger$.

## 2.4 Modified index

The fact that the contracted large $\mathcal{N} = 4$ superconformal algebra contains 4 pairs of free bosons and fermions suggests that any theory enjoying this symmetry contains a copy of the

free $T^4$ sigma model. The presence of free fermions then renders the standard supersymmetric indices vanishing. To obtain meaningful indices, one way is to decouple the free $T^4$ sigma model and study the quotient theory, which has a different symmetry algebra (see [62, 63] for discussions in a related model). Instead, we follow the classic treatment [28] to define and compute a modified index.[5]

The BPS states satisfying (2.15) are in the one-dimensional short multiplet of the super-symmetry algebra (2.14), while the non-BPS states form four-dimensional long multiplets. Hence, we can construct an index that counts the number of short multiplets. Let us start with the NS sector partition function (with $(-1)^F$ inserted in the trace),

$$Z_{\text{NS}}(\tau, \bar{\tau}, z, \bar{z}) = \text{Tr}_{\text{NS}} \left[ (-1)^F q^{L_0 - \frac{c}{24}} \bar{q}^{\widetilde{L}_0 - \frac{c}{24}} y^{2K_0^3} \bar{y}^{2\widetilde{K}_0^3} \right] , \qquad (2.33)$$

where $y = e^{2\pi i z}$. The index is defined by

$$I'_{\text{NS}}(\tau, z) = \bar{q}^{\frac{c}{24}} Z_{\text{NS}}(\tau, \bar{\tau}, z, \bar{z}) \big|_{\bar{z} = -\frac{1}{2}\bar{\tau}} = \text{Tr}_{\text{NS}} \left[ (-1)^F q^{L_0 - \frac{c}{24}} y^{2K_0^3} \right] . \qquad (2.34)$$

However, because the BPS states form quartets under the Clifford algebra (2.16) from the free fermions $\psi^\pm$ and $\bar{\psi}^\pm$, the index $I'_{\text{NS}}$ always vanishes due to the cancellation between bosons and fermions.

To get a non-vanishing counting, we consider the modified index

$$\begin{aligned} I_{\text{NS}}(\tau, z) &= \frac{1}{2} \text{Tr}_{\text{NS}} \left[ (-1)^F \left( 2\widetilde{K}_0^3 \right) \left( 2\widetilde{K}_0^3 - 1 \right) q^{L_0 - \frac{c}{24}} y^{2K_0^3} \right] \\ &= \frac{1}{2} \bar{q}^{\frac{c}{24} - 1} \partial_{\bar{y}}^2 Z_{\text{NS}}(\tau, \bar{\tau}, z, \bar{z}) \big|_{\bar{z} = -\frac{1}{2}\bar{\tau}} , \end{aligned} \qquad (2.35)$$

which is related to the modified index $I_R(\tau, z)$ in [28] by a spectral flow from the NS-sector to the Ramond (R) sector,

$$I_R(\tau, z) = q^{\frac{c}{24}} y^{-\frac{c}{6}} I_{\text{NS}} \left( \tau, z - \frac{\tau}{2} \right) . \qquad (2.36)$$

The modified index in the R-sector can be computed by taking derivatives on the R-sector partition function (with $(-1)^F$ inserted in the trace) as

$$\begin{aligned} I_R(\tau, z) &= \frac{1}{2} \partial_{\bar{y}}^2 Z_R(\tau, \bar{\tau}, z, \bar{z}) \big|_{\bar{z} = 0} , \\ Z_R(\tau, \bar{\tau}, z, \bar{z}) &= \text{Tr}_R \left[ (-1)^F q^{L_0 - \frac{c}{24}} \bar{q}^{\widetilde{L}_0 - \frac{c}{24}} y^{2J_0} \bar{y}^{2\widetilde{J}_0} \right] . \end{aligned} \qquad (2.37)$$

---

[5]Performing the $T^4$ quotient is akin to studying the 4d $\mathcal{N} = 4$ super-Yang-Mills with gauge group $SU(N)$ instead of $U(N)$, by factoring out the "center-of-mass" $U(1)$ free theory.

The spectral flow acts on the R- and NS-sector partition functions as

$$Z_{\mathrm{R}}(\tau, \bar{\tau}, z, \bar{z}) = q^{\frac{c}{24}} y^{-\frac{c}{6}} \bar{q}^{\frac{c}{24}} \bar{y}^{-\frac{c}{6}} Z_{\mathrm{NS}}\left(\tau, \bar{\tau}, z - \frac{\tau}{2}, \bar{z} - \frac{\bar{\tau}}{2}\right). \tag{2.38}$$

Following [28], let us compute the modified index in the $T^4$ symmetric orbifold theory. We start with the partition function of the $T^4$ CFT in the R-sector

$$Z_{T^4,\mathrm{R}} = \left(\frac{\theta_1(z|\tau)}{\eta(\tau)}\right)^2 \frac{1}{\eta(\tau)^4} \overline{\left(\frac{\theta_1(z|\tau)}{\eta(\tau)}\right)^2 \frac{1}{\eta(\tau)^4}}, \tag{2.39}$$

where we have restricted to the zero momentum and winding sector. Let us rewrite the partition function as

$$Z_{T^4,\mathrm{R}} = \sum_{h,\tilde{h},j,\tilde{j}} c(h, \tilde{h}, 2j, 2\tilde{j}) q^h \bar{q}^{\tilde{h}} y^{2j} \bar{y}^{2\tilde{j}}, \tag{2.40}$$

where the coefficients $c(h, \tilde{h}, j, \tilde{j})$ can be extracted from the explicit formula (2.39). Now, let us consider the index

$$I_{T^4,\mathrm{R}} = \frac{1}{2}\partial_{\bar{y}}^2 Z_{T^4,\mathrm{R}}\big|_{\bar{y}=1} = -\left(\frac{\theta_1(z|\tau)}{\eta(\tau)}\right)^2 \frac{1}{\eta(\tau)^4} = \sum_{h,j} \hat{c}(h, 2j) q^h y^{2j}, \tag{2.41}$$

where the coefficients $\hat{c}(h, 2j)$ and $c(h, \tilde{h}, 2j, 2\tilde{j})$ are related by

$$\hat{c}(h, 2j) = \frac{1}{2}\sum_{\tilde{j}} (2\tilde{j})^2 c(h, 0, 2j, 2\tilde{j}). \tag{2.42}$$

By the Dijkgraaf-Moore-Verlinde-Verlinde (DMVV) formula [64], the grand partition function of the $T^4$ symmetric orbifold theories is given by

$$\mathcal{Z}_{\mathrm{R}} = \sum_{k=0}^{\infty} p^k Z_{\mathrm{Sym}^k(T^4),\mathrm{R}} = \prod_{n=1}^{\infty} \prod_{\substack{h,\tilde{h},j,\tilde{j} \\ h-\tilde{h}\in n\mathbb{Z}}} \frac{1}{(1 - p^n q^{\frac{h}{n}} \bar{q}^{\frac{\tilde{h}}{n}} y^{2j} \tilde{y}^{2\tilde{j}})^{c(h,\tilde{h},2j,2\tilde{j})}}, \tag{2.43}$$

where $c(h, \tilde{h}, 2j, 2\tilde{j})$ is the coefficient in the single-copy partition function (2.40). Now, let us compute the grand index

$$\mathcal{I}_{\mathrm{R}} = \sum_{k=0}^{\infty} p^k I_{\mathrm{Sym}^k(T^4),\mathrm{R}} = \frac{1}{2}\partial_{\bar{y}}^2 \mathcal{Z}_{\mathrm{R}}\big|_{\bar{y}=1} = \sum_{n=1}^{\infty} \sum_{m=0}^{\infty} \sum_{j\in\frac{1}{2}\mathbb{Z}} \frac{\hat{c}(nm, 2j) p^n q^m y^{2j}}{(1 - p^n q^m y^{2j})^2}. \tag{2.44}$$

The index in the NS sector can be obtained by the spectral flow from the R sector

$$\mathcal{I}_{\mathrm{NS}}(p,q,y) = \sum_{k=0}^{\infty} p^k I_{\mathrm{Sym}^k(T^4),\mathrm{NS}} = \mathcal{I}_{\mathrm{R}}(pq^{\frac{1}{4}}y, q, q^{\frac{1}{2}}y) \,. \tag{2.45}$$

For instance, the expansion of the $N = 2$ and $3$ indices are

$$\begin{aligned}
I_{\mathrm{NS}}^{\mathrm{Sym}^2(T^4)} &= \frac{2}{q^{\frac{1}{2}}} + \left(y + \frac{1}{y}\right) - q^{\frac{1}{2}}\left(6y^2 + 12 + \frac{6}{y^2}\right) + O(q) \,, \\
I_{\mathrm{NS}}^{\mathrm{Sym}^3(T^4)} &= \frac{3}{q^{\frac{3}{4}}} + q^{\frac{1}{4}}\left(y^2 + 8 + \frac{1}{y^2}\right) - 8q^{\frac{3}{4}}\left(y^3 + 7y + \frac{7}{y} + \frac{1}{y^3}\right) + O\left(q^{\frac{5}{4}}\right) \,.
\end{aligned} \tag{2.46}$$

We give the expansions of the $N = 2$ and $3$ indices up to $q^3$ in Appendix B.

## 2.5 Exact BPS partition function at $N = 2$

For $N = 2$, the index actually contains the information on the exact degeneracy of $\frac{1}{4}$-BPS states [49,41]. At central charge $c = 12$, the contracted large $\mathcal{N} = 4$ superconformal algebra has two short multiplets with $j = 0, \frac{1}{2}$ and the NS-sector characters $\chi_{2j}$ and a single long mulitplet with $j = 0$, $h \geq 0$ and the NS-sector character $\chi_{h,j}$. The two short multiplets combine into a long multiplet as

$$\chi_0 + 2\chi_1 = \chi_{0,0} \,. \tag{2.47}$$

In Appendix B, we give the expansions of the characters $\chi_0$ and $\chi_1$.

It is reasonable to assume that at a generic point in the moduli space, there are no additional conserved currents [49].[6] We can then write down a formula of the partition function of $\frac{1}{4}$-BPS states (satisfying (2.15)),

$$Z_{N=2}^{\mathrm{BPS}} = n_0 \chi_0 \overline{\chi_0^{\mathrm{BPS}}} + n_1 \chi_1 \overline{\chi_1^{\mathrm{BPS}}} + \sum_{h=1}^{\infty} N_h q^h (\chi_0 + 2\chi_1) \overline{\chi_1^{\mathrm{BPS}}} \,, \tag{2.48}$$

where $\overline{\chi_{j,\mathrm{BPS}}}$ consists of the terms in $\overline{\chi_j}$ satisfying the BPS condition, i.e. of the form $\bar{y}(\bar{q}^{\frac{1}{2}}\bar{y})^m$ for $m \in \mathbb{Z}$,

$$\overline{\chi_0^{\mathrm{BPS}}} := \bar{q}^{\frac{3}{2}}\bar{y}^4 - 2\bar{q}\bar{y}^3 + 2\sqrt{\bar{q}}\bar{y}^2 + \frac{1}{\sqrt{\bar{q}}} - 2\bar{y}, \quad \overline{\chi_1^{\mathrm{BPS}}} := -\bar{q}\bar{y}^3 + 2\sqrt{\bar{q}}\bar{y}^2 - \bar{y} \,. \tag{2.49}$$

---

[6]We thank Nathan Benjamin for a discussion on this point.

Taking $\bar{y}$-derivatives, we find

$$I^{\mathrm{Sym}^2(T^4)}_{\mathrm{NS}} = \frac{1}{2}\bar{q}^{\frac{c}{24}-1}\partial^2_{\bar{y}}Z^{\mathrm{BPS}}_{N=2}\big|_{\bar{y}=\bar{q}^{-\frac{1}{2}}} = 2n_0\chi_0 - n_1\chi_1 - \sum_{h=1}^{\infty}N_h q^h(\chi_0 + 2\chi_1)\,. \tag{2.50}$$

The numbers $n_0$, $n_1$ and $N_h$ can be determined by expanding the modified index given in (2.44), (2.45) in terms of the characters. For instance, using the expansions for the index $I^{\mathrm{Sym}^2(T^4)}_{\mathrm{NS}}$ and the characters $\chi_0$ and $\chi_1$ given in (B.1), (B.3), and (B.4) in Appendix B, we find $n_0 = 2$, $n_1 = 5$, $N_2 = 42$, $N_3 = 70$, and $N_4 = 324$. The BPS partition function can be expressed in terms of the modified index as

$$Z^{(N=2)}_{\mathrm{BPS}} = \mathcal{S}_0\overline{\chi^{\mathrm{BPS}}_0} + \mathcal{S}_1\overline{\chi^{\mathrm{BPS}}_1}\,, \quad \mathcal{S}_0 := \chi_0\,, \quad \mathcal{S}_1 := 2\chi_0 - I^{\mathrm{Sym}^2(T^4)}_{\mathrm{NS}} \tag{2.51}$$

The expansions of $\mathcal{S}_0$ and $\mathcal{S}_1$ are

$$\begin{aligned}
\mathcal{S}_0 =&\frac{1}{q^{\frac{1}{2}}} - 2\left(y + \frac{1}{y}\right) + q^{\frac{1}{2}}\left(2y^2 + \frac{2}{y^2} + 9\right) - 2q\left(y^3 + 9y + \frac{9}{y} + \frac{1}{y^3}\right) + O\left(q^{\frac{3}{2}}\right)\,, \\
\mathcal{S}_1 =&-5\left(y + \frac{1}{y}\right) + 10q^{\frac{1}{2}}\left(y^2 + 3 + \frac{1}{y^2}\right) - 5q\left(y^3 + 15y + \frac{15}{y} + \frac{1}{y^3}\right) + O\left(q^{\frac{3}{2}}\right)\,.
\end{aligned} \tag{2.52}$$

We give the expansions of $\mathcal{S}_0 = \chi_0$ and $\mathcal{S}_1$ up to $q^3$ in (B.3) and (B.5) in Appendix B. In Section 3.3, we will compare this partition function with the result from the computation of the $Q$-cohomology.

# 3 Lifting as a supercharge cohomology problem

In Section 2.3 we reviewed the lifting problem of $\frac{1}{4}$-BPS states under second-order conformal perturbation theory around the symmetric orbifold point. In particular, we saw in (2.32) that the anomalous dimension is dictated by the $Q, Q^{\dagger}$ action with first-order conformal perturbation, $\delta\tilde{h} = \frac{1}{2}\{Q^{\dagger}, Q\}$. Noting that the right hand side is of the form of a Laplacian, we can leverage the standard Hodge theory framework and establish that the space of unlifted states under second-order perturbation theory is isomorphic to the cohomology of the supercharge $Q$ computed within first-order conformal perturbation theory. This correspondence allows us to systematically identify protected $\frac{1}{4}$-BPS operators by analyzing the supercharge cohomology.

In the remainder of this section, we evaluate the supercharge cohomology at finite $N$. At the orbifold point, operators are characterized by various cycle shapes. In the general analysis, we anticipate that operators with different cycle shapes may mix due to the action

of the first-order deformed supercharge.

## 3.1  Holographic covering Hilbert space

We introduce a "second-quantized" basis for the full Hilbert space (2.7), which then leads to a natural holographic cover [7].[7] We define a *trivial cycle* to be a length-one cycle in its vacuum state $|1\rangle$, and define

$$\mathcal{H}'_{(1)} := (1 - |1\rangle\langle 1|)\,\mathcal{H}_{(1)} \tag{3.1}$$

to be the Hilbert space of excited states in $\mathcal{H}_{(1)}$. Applying the isomorphism (2.8), and decomposing $\mathcal{H}_{(1)} = |1\rangle \oplus \mathcal{H}'_{(1)}$, the symmetric product of length-one cycles can be expressed as[8]

$$S^{N_1}\mathcal{H}_{(1)} \cong \bigoplus_{N'_1=0}^{N_1} |1\rangle^{\otimes N_1 - N'_1} \otimes S^{N'_1}\mathcal{H}'_{(1)}. \tag{3.2}$$

When $N_1 = N$, this describes the untwisted sector of the symmetric product theory.

Omitting the trivial cycles, the full Hilbert space can be written as

$$\mathcal{H}_N \cong \bigoplus_{\substack{N_1, N_2, N_3, \cdots \\ N \geq N_1 + 2N_2 + 3N_3 + \cdots}} S^{N_1}\mathcal{H}'_{(1)} \bigotimes_{n=2}^{\infty} S^{N_n}\mathcal{H}^{\mathbb{Z}_n}_{(n)}. \tag{3.3}$$

Given $(N_1, N_2, N_3, \cdots)$, we define the *number of nontrivial cycles* $\nu$ and the *total length of nontrivial cycles* $\ell$ as

$$\nu(N_n) := \sum_n N_n, \quad \ell(N_n) := \sum_n nN_n. \tag{3.4}$$

As will be useful momentarily, we can assemble the orthogonal projectors $P^{(N_1, N_2, N_3, \cdots)}$ into orthogonal projectors onto definite $(\nu, \ell)$,

$$P^{(\nu, \ell)} := \sum_{\substack{N_1, N_2, N_3, \cdots \\ \nu(N_n)=\nu,\ \ell(N_n)=\ell}} P^{(N_1, N_2, N_3, \cdots)}. \tag{3.5}$$

The decomposition (3.3) also allows us to define an infinite $N$ holographic covering Hilbert

---

[7]This is not to be confused with the notion of second-quantization in [64], which can be viewed as taking a direct sum of $\mathcal{H}_N$ over all $N$.

[8]Note that $N'_1$ is not a quantum number, i.e. there does not exist a linear operator that measures $N'_1$. This follows from the fact that $N'_1$ is not a grading of the vector space since it does not respect addition. Given an arbitrary $|\psi\rangle \in \mathcal{H}'_{(1)}$, the states $|\pm\rangle := |1\rangle \pm |\psi\rangle$ have $N'_1 = 1$, but $|+\rangle - |-\rangle$ has $N'_1 = 0$. This comment also applies to the $N_1, N_2, N_3, \cdots$ below.

space as

$$\mathcal{H}_\infty \cong \bigoplus_{N_1,N_2,N_3,\cdots} S^{N_1}\mathcal{H}'_{(1)} \bigotimes_{n=2}^{\infty} S^{N_n}\mathcal{H}^{\mathbb{Z}_n}_{(n)}. \tag{3.6}$$

The finite $N$ Hilbert space $\mathcal{H}_N$ can be viewed as a quotient of the infinite $N$ Hilbert space $\mathcal{H}_\infty$ by the subspace consisting of states with a total length of non-trivial cycles exceeding $N$. This quotient is known as the "stringy exclusion principle" [27] in the literature.

## 3.2   Deformed $Q$-action on the free BPS sector

Under first-order conformal perturbation theory, the computation of the supercharge $Q$-action involves correlators with a single insertion of the deformation operator (2.31) which is in the length-2 single-cycle sector. According to the symmetric group product law, the $Q$-action can change the cycle shapes in two processes: *joining* two cycles into one and *splitting* one cycle into two. Each process can be further divided into two cases depending on whether a trivial cycle is involved or not. And as we will see, the $Q$-action on a general cycle shape is given by a sum over all pairs of cycles, so it suffices to study the $Q$-action on a state with only one or two nontrivial cycles. In the following, $Q$ always represents the first-order deformed supercharge, though we often call it the supercharge for simplicity.

Since the anomalous dimensions $\delta\tilde{h}$ are infinitesimal in the conformal perturbation theory, the lifting of BPS states can be studied in the free BPS sector, which consists of all the $\frac{1}{4}$-BPS states (BPS on the right) at the free orbifold point. In this sector, the supercharge $Q$ has a decomposition

$$Q = Q^{(0,1)} + Q^{(-1,0)} + Q^{(1,0)} \tag{3.7}$$

where $Q^{(\nu,\ell)}$ maps a sector with $\nu_0$ non-trivial cycles and total non-trivial cycle length $\ell_0$ to a new sector with $\nu_0 + \nu$ non-trivial cycles and total non-trivial cycle length $\ell_0 + \ell$. In terms of the orthogonal projectors defined in (3.5),

$$Q^{(\nu,\ell)} := \sum_{(\nu_0,\ell_0)} P^{(\nu_0+\nu,\ell_0+\ell)} Q P^{(\nu_0,\ell_0)}. \tag{3.8}$$

Let us depict their actions on a Clifford quartet while keeping track of the right-moving R-charges and recalling that $Q$ carries $j = \tilde{j} = \frac{1}{2}$. The $Q^{(0,1)}$-action joins a non-trivial cycle

with a trivial cycle,

$$
\tag{3.9}
$$

$Q^{(-1,0)}$ joins two nontrivial cycles,

$$
\tag{3.10}
$$

and $Q^{(1,0)}$ splits a non-trivial cycle into two non-trivial cycles,

$$
\tag{3.11}
$$

The naive $Q^{(0,-1)}$ for

$$
\tag{3.12}
$$

does not exist due to the right-moving R-charge conservation. We will loosely refer to the maps $Q^{(\nu,\ell)}$ as supercharges. More explicitly, their actions take the forms

$$
\begin{aligned}
Q^{(0,1)} \left| \{w_i\} \right\rangle &= \sum_j c^{(0,1)}_{w;i,j} \left| \{(w+1)_j\} \right\rangle, \\
Q^{(-1,0)} \left| \{n_{i_1}, (w-n)_{i_2}\} \right\rangle &= \sum_j c^{(-1,0)}_{n,w-n;i_1,i_2,j} \left| \{w_j\} \right\rangle, \\
Q^{(1,0)} \left| \{w_i\} \right\rangle &= \sum_{n=1}^{w-1} \sum_{j_1,j_2} c^{(1,0)}_{n,w-n;i,j_1 j_2} \left| \{n_{j_1}, (w-n)_{j_2}\} \right\rangle,
\end{aligned}
\tag{3.13}
$$

where we write the states using the convention in (2.23), and for simplicity, we have omitted the $\alpha$, $\beta$ indices of the states. The constraints from the Clifford algebra (2.16) on the $Q$-

action will be studied in Section 3.3 and Section 5.1.

The nilpotency of $Q$ implies the nilpotency and mutual anti-commutativity of $Q^{(\nu,\ell)}$,

$$\{Q^{(\nu,\ell)}, Q^{(\nu',\ell')}\} = 0\,, \tag{3.14}$$

which can be easily seen by applying (3.8). Thus, each $Q^{(\nu,\ell)}$ can be treated as a differential on a cochain complex, thereby defining a cohomology. In the following, we present our explicit computation of the $Q^{(\nu,\ell)}$ action, largely borrowing but further extending the technology developed in [50]. More precisely, the formulae laid out in [50] is the case of $Q^{(0,1)}$, which dominates in the large $N$ limit as will be explained in Section 3.4.

To determine the coefficients $c^{(\nu,\ell)}$, we evaluate the matrix elements using conformal perturbation theory. For illustration, we review the procedure to obtain $c_{w;i,j}^{(0,1)}$ in detail and briefly outline how to compute the remaining three coefficients at the end of this subsection. We have

$$
\begin{aligned}
c_{w;i,j}^{(0,1)} &= \frac{\langle\{(w+1)_j\}|\, Q\,|\{w_i\}\rangle}{\langle\{(w+1)_j\}|\{(w+1)_j\}\rangle} \\
&= (w+1)\,\langle(w+1)_j|\, Q\,|w_i\rangle \\
&= (w+1)\oint_{\bar{x}=0} d\bar{x}\int d^2y\,\langle(w+1)_j|\,\tilde{G}^+(\bar{x})\Phi(y,\bar{y})\,|w_i\rangle\,,
\end{aligned}
\tag{3.15}
$$

where $\tilde{G}^+(\bar{x})$ is the right-moving supercharge current, and $\Phi(y,\bar{y})$ is the exactly marginal deformation operator inserted at position $(y,\bar{y})$. From the first to the second line in (3.15), we apply the formula (2.5), and choose the cycles of the states $|w_i\rangle$, $|(w+1)_j\rangle$ and the supercharge $Q$ to be $(1,2,\cdots,w)$, $(1,2,\cdots,w+1)$ and $(w,w+1)$, respectively.

Following the approach outlined in [50], taking the OPE between the supercurrent and the deformation operator removes the right-moving supercharge $\tilde{G}_{-\frac{1}{2}}$ from the deformation operator. After completing the integrals, the matrix element $\langle(w+1)_j|\, Q\,|w_i\rangle$ is proportional to the three-point function coefficient

$$\langle(w+1)_j|\, V(G'^{-}_{-\frac{1}{2}}\,|2_{--,--}\rangle)(1)\,|w_i\rangle \tag{3.16}$$

where $V(G'^{-}_{-\frac{1}{2}}\,|2_{--,--}\rangle)(1)$ represents the operator corresponding to the state $G'^{-}_{-\frac{1}{2}}\,|2_{--,--}\rangle$, inserted at $z=1$.

The calculation of (3.16) can be significantly simplified by mapping the computation to the covering space, which we parameterize as the $t$-plane, using a covering map

$$z(t) = \Gamma(t) = (w+1)t^w - wt^{w+1}\,. \tag{3.17}$$

This transformation eliminates the branch cuts in the $z$-plane and hence converts the fractional bosonic and fermionic modes into integer or half-integer modes in the covering space. For instance, the fractional modes are mapped as follows:

$$\alpha_{\frac{m}{w}} = \oint_{z=0} \partial X^{[m]}(z)\, z^{\frac{m}{w}}\, dz \quad \longrightarrow \quad \oint_{t=0} \partial X(t)\, \Gamma(t)^{\frac{m}{w}}\, dt\,,$$

$$\psi_{\frac{m}{w}} = \oint_{z=0} \psi^{[m]}(z)\, z^{\frac{m}{w}-\frac{1}{2}}\, dz \quad \longrightarrow \quad \oint_{t=0} \psi(t) \left(\frac{d\Gamma(t)}{dt}\right)^{\frac{1}{2}} \Gamma(t)^{\frac{m}{w}-\frac{1}{2}}\, dt\,. \tag{3.18}$$

where $\partial X^{[m]}$ and $\psi^{[m]}$ for $m = 0, \cdots, w-1$ on the $z$-plane are defined in (2.9) and are lifted to single-valued fields $\partial X$ and $\psi$ on the $t$-plane. Upon performing a series expansion around $t = 0$, the factors introduced by the covering map $\Gamma(t)$ yield only integer or half-integer powers of $t$.

Using the lifting formula (3.18), the computation of (3.16) reduces to contour integrals on the $t$-plane, with the integrands being products of the following terms:

- Covering map factors: These factors arise from the conformal transformation induced by the covering map $z = \Gamma(t)$. Their specific form is dictated by the choice of the covering map determined by the cycle shapes of the initial and final states, the left-moving weights of the modes acting on these states, and the fermionic or bosonic nature of the modes.

- Correlation functions: Both the $\mathbb{Z}_2$ twisted operator inserted at $z = 1$ and the chiral-chiral primary states involved in $|w_i\rangle$ and $|(w+1)_j\rangle$ in (3.16) are lifted to spin fields on the covering space (see Appendix A). Hence, we get a product of a free boson correlator and a correlation function of free fermions with spin fields. The former can be simply computed using Wick contraction, while the latter involving spin fields and fermions can be computed using bosonization techniques with formulae given in the appendix.

The coefficients $c^{(1,0)}_{n,w-n;i,j_1j_2}$ and $c^{(-1,0)}_{n,w-n;i_1,i_2,j}$ can be derived in a similar manner. The symmetry factor is determined by utilizing formula (2.5). An additional complication in evaluating $c^{(\pm 1,0)}_{n,w-n;ijk}$ compared to $c^{(0,1)}_{w;ij}$ is that in these cases, a three-point function is mapped to a four-point function in the covering space. Consequently, it is necessary to include contours around the fourth point in the covering space. In Appendix C, we present an example that demonstrates the the nilpotency of $Q^{(0,1)}$ action.

Finally, let us consider $\widetilde{G}^-_{\frac{1}{2}} = Q^\dagger$, the Hermitian conjugate of the first-order deformed conformal supercharge. It admits the decomposition

$$Q^\dagger = (Q^\dagger)^{(0,-1)} + (Q^\dagger)^{(1,0)} + (Q^\dagger)^{(-1,0)}\,, \tag{3.19}$$

| supercharge | covering map | symmetry factor | Leibniz | conjugate | Leibniz |
|---|---|---|---|---|---|
| $Q^{(0,1)}$ | $(w+1)t^w - wt^{w+1}$ | $w+1$ | Y | $(Q^\dagger)^{(0,-1)}$ | Y |
| $Q^{(-1,0)}$ | $t^{w-n}(t - \frac{w}{w-n})^n$ | $w$ | N | $(Q^\dagger)^{(1,0)}$ | Y |
| $Q^{(1,0)}$ | $t^w(t - \frac{w-n}{w})^{-n}$ | $(w-n)n$ | Y | $(Q^\dagger)^{(-1,0)}$ | N |

Table 1: The covering maps and symmetry factors for the supercharges $Q^{(\nu,\ell)}$ and their conjugates, and whether their actions satisfy the Leibniz rule when acting on multi-cycle states.

where $(Q^\dagger)^{(\nu,\ell)}$ are defined by

$$(Q^\dagger)^{(\nu,\ell)} := \sum_{(\nu_0,\ell_0)} P^{(\nu_0+\nu,\ell_0+\ell)} Q^\dagger P^{(\nu_0,\ell_0)}. \tag{3.20}$$

The matrix elements of $(Q^\dagger)^{(\nu,\ell)}$ can be computed following a similar procedure as the one described above for $Q^{(\nu,\ell)}$. Alternatively, one can use the fact that the first-order deformed $Q^\dagger$ can be obtained from the first order deformed $Q$ by taking the Hermitian conjugate $\dagger$ in the free orbifold theory. Such a Hermitian conjugate respects the total number $\nu$ and total length $\ell$ of the non-trivial cycles; hence, we have

$$(Q^\dagger)^{(\nu,\ell)} = Q^{(-\nu,-\ell)\dagger}. \tag{3.21}$$

From their explicit actions it is easy to see that $Q^{(0,1)}, Q^{(1,0)}, (Q^\dagger)^{(0,-1)}, (Q^\dagger)^{(1,0)}$ satisfy the Leibniz rule when acting on multi-cycle states, while $Q^{(0,1)}, (Q^\dagger)^{(-1,0)}$ do not.

The key properties of these supercharges discussed above are summarized in Table 1.

## 3.3 Explicit cohomology representatives at $N = 2$

To illustrate the procedure outlined in the previous subsection, we further study the case of $N = 2$. There are only two distinct cycle shapes, $(1,1)$ and $(2)$. The $\frac{1}{2}$-BPS states are

$$|\{1_{\pm\pm,\pm\pm}, 1_{\pm\pm,\pm\pm}\}\rangle, \quad |2_{\pm\pm,\pm\pm}\rangle, \tag{3.22}$$

and we represent the candidate $\frac{1}{4}$-BPS states as

$$|\{1_{i_1,\pm\pm}, 1_{i_2,\pm\pm}\}\rangle, \quad |2_{j,\pm\pm}\rangle, \tag{3.23}$$

where $i$, $i_2$, $j$ denote all possible left-moving modes.

The supercharge $Q$ can map a state in $V_{(1,1)}$ to $V_{(2)}$ through the joining process, or map a state in $V_{(2)}$ to $V_{(1,1)}$ through splitting. The right-moving Clifford algebra (2.16) imposes selection rules that further constrain the $Q$-action. Since only the diagonal symmetry algebra is preserved under the exactly marginal deformation, we first decompose the states $|\{1_{i_1,\pm\pm}, 1_{i_2,\pm\pm}\}\rangle$ into four quartets of the diagonal right-moving Clifford algebra, each an eigenvector of the exchange symmetry:

$$
V_{(1,1)S_1} := \mathrm{span}
\begin{bmatrix}
(\tilde{\psi}^{(1)+}_{-\frac{1}{2}} + \tilde{\psi}^{(2)+}_{-\frac{1}{2}})(\tilde{\bar{\psi}}^{(1)+}_{-\frac{1}{2}} + \tilde{\bar{\psi}}^{(2)+}_{-\frac{1}{2}}) |\Psi_{(ij)}\rangle \\
(\tilde{\bar{\psi}}^{(1)+}_{-\frac{1}{2}} + \tilde{\bar{\psi}}^{(2)+}_{-\frac{1}{2}}) |\Psi_{(ij)}\rangle \\
(\tilde{\psi}^{(1)+}_{-\frac{1}{2}} + \tilde{\psi}^{(2)+}_{-\frac{1}{2}}) |\Psi_{(ij)}\rangle \\
|\Psi_{(ij)}\rangle
\end{bmatrix}
, \quad
\tilde{j} =
\begin{bmatrix}
1 \\
\frac{1}{2} \\
\frac{1}{2} \\
0
\end{bmatrix}
\tag{3.24}
$$

$$
V_{(1,1)S_2} := \mathrm{span}
\begin{bmatrix}
\tilde{\psi}^{(1)+}_{-\frac{1}{2}} \tilde{\psi}^{(2)+}_{-\frac{1}{2}} \tilde{\bar{\psi}}^{(1)+}_{-\frac{1}{2}} \tilde{\bar{\psi}}^{(2)+}_{-\frac{1}{2}} |\Psi_{(ij)}\rangle \\
\tilde{\psi}^{(1)+}_{-\frac{1}{2}} \tilde{\psi}^{(2)+}_{-\frac{1}{2}} (\tilde{\bar{\psi}}^{(1)+}_{-\frac{1}{2}} - \tilde{\bar{\psi}}^{(2)+}_{-\frac{1}{2}}) |\Psi_{(ij)}\rangle \\
(\tilde{\psi}^{(1)+}_{-\frac{1}{2}} - \tilde{\psi}^{(2)+}_{-\frac{1}{2}}) \tilde{\bar{\psi}}^{(1)+}_{-\frac{1}{2}} \tilde{\bar{\psi}}^{(2)+}_{-\frac{1}{2}} |\Psi_{(ij)}\rangle \\
(\tilde{\psi}^{(1)+}_{-\frac{1}{2}} - \tilde{\psi}^{(2)+}_{-\frac{1}{2}})(\tilde{\bar{\psi}}^{(1)+}_{-\frac{1}{2}} - \tilde{\bar{\psi}}^{(2)+}_{-\frac{1}{2}}) |\Psi_{(ij)}\rangle
\end{bmatrix}
, \quad
\tilde{j} =
\begin{bmatrix}
2 \\
\frac{3}{2} \\
\frac{3}{2} \\
1
\end{bmatrix}
, \tag{3.25}
$$

$$
V_{(1,1)A_1} := \mathrm{span}
\begin{bmatrix}
\tilde{\psi}^{(1)+}_{-\frac{1}{2}} \tilde{\psi}^{(2)+}_{-\frac{1}{2}} (\tilde{\bar{\psi}}^{(1)+}_{-\frac{1}{2}} + \tilde{\bar{\psi}}^{(2)+}_{-\frac{1}{2}}) |\Psi_{[ij]}\rangle \\
(\tilde{\psi}^{(1)+}_{-\frac{1}{2}} - \tilde{\psi}^{(2)+}_{-\frac{1}{2}})(\tilde{\bar{\psi}}^{(1)+}_{-\frac{1}{2}} + \tilde{\bar{\psi}}^{(2)+}_{-\frac{1}{2}}) |\Psi_{[ij]}\rangle \\
\tilde{\psi}^{(1)+}_{-\frac{1}{2}} \tilde{\psi}^{(2)+}_{-\frac{1}{2}} |\Psi_{[ij]}\rangle \\
(\tilde{\psi}^{(1)+}_{-\frac{1}{2}} - \tilde{\psi}^{(2)+}_{-\frac{1}{2}}) |\Psi_{[ij]}\rangle ,
\end{bmatrix}
, \quad
\tilde{j} =
\begin{bmatrix}
\frac{3}{2} \\
1 \\
1 \\
\frac{1}{2}
\end{bmatrix}
\tag{3.26}
$$

$$
V_{(1,1)A_2} := \mathrm{span}
\begin{bmatrix}
\tilde{\bar{\psi}}^{(1)+}_{-\frac{1}{2}} \tilde{\bar{\psi}}^{(2)+}_{-\frac{1}{2}} (\tilde{\psi}^{(1)+}_{-\frac{1}{2}} + \tilde{\psi}^{(2)+}_{-\frac{1}{2}}) |\Psi_{[ij]}\rangle \\
(\tilde{\bar{\psi}}^{(1)+}_{-\frac{1}{2}} - \tilde{\bar{\psi}}^{(2)+}_{-\frac{1}{2}})(\tilde{\psi}^{(1)+}_{-\frac{1}{2}} + \tilde{\psi}^{(2)+}_{-\frac{1}{2}}) |\Psi_{[ij]}\rangle \\
\tilde{\bar{\psi}}^{(1)+}_{-\frac{1}{2}} \tilde{\bar{\psi}}^{(2)+}_{-\frac{1}{2}} |\Psi_{[ij]}\rangle \\
(\tilde{\bar{\psi}}^{(1)+}_{-\frac{1}{2}} - \tilde{\bar{\psi}}^{(2)+}_{-\frac{1}{2}}) |\Psi_{[ij]}\rangle
\end{bmatrix}
, \quad
\tilde{j} =
\begin{bmatrix}
\frac{3}{2} \\
1 \\
1 \\
\frac{1}{2}
\end{bmatrix}
, \tag{3.27}
$$

where $|\Psi_{ij}\rangle := |1_{i,--}, 1_{j,--}\rangle$ and $(ij)$, $[ij]$ denotes symmetrization and antisymmetrization.

Recall the following facts:

1. The supercharge $Q$ carries right-moving R-charge $\tilde{j} = \frac{1}{2}$;

2. $Q$ commutes with the right-moving Clifford algebra in the free BPS sector;

3. The right-moving R-charge of a general multi-cycle state (2.23) in the free BPS sector with definite $\nu$ and $\ell$ is bounded (2.29) as

$$\frac{\ell - \nu}{2} \leq \tilde{j} \leq \frac{\ell + \nu}{2}. \tag{3.28}$$

For cycle shape $(1,1)$ this is

$$0 \leq \tilde{j} \leq 2, \tag{3.29}$$

and for cycle shape $(2)$ this is

$$\frac{1}{2} \leq \tilde{j} \leq \frac{3}{2}. \tag{3.30}$$

Now, consider $Q$ acting on states in $V_{(1,1)A_2}$, which must give a state in $V_{(2)}$. Since $Q$ acting on the top component gives a state with $\tilde{j} = \frac{3}{2} + \frac{1}{2} = 2$ violating (3.30), the $Q$-action must annihilate the states in $V_{(1,1)A_2}$. Repeatedly applying similar arguments, we arrive at the following cochain complexes:

$$0 \xrightarrow{Q} V_{(1,1)S_1} \xrightarrow{Q} V_{(2)} \xrightarrow{Q} V_{(1,1)S_2} \xrightarrow{Q} 0\,,$$
$$0 \xrightarrow{Q} V_{(1,1)A_1} \xrightarrow{Q} 0\,, \tag{3.31}$$
$$0 \xrightarrow{Q} V_{(1,1)A_2} \xrightarrow{Q} 0\,.$$

We observe that the states in $V_{(1,1)A_1}$ and $V_{(1,1)A_2}$ must represent non-trivial supercharge cohomology classes.

Applying similar arguments to the $Q^\dagger$-action, we obtain the following complexes:

$$0 \xrightarrow{Q^\dagger} V_{(1,1)S_2} \xrightarrow{Q^\dagger} V_{(2)} \xrightarrow{Q^\dagger} V_{(1,1)S_1} \xrightarrow{Q^\dagger} 0\,,$$
$$0 \xrightarrow{Q^\dagger} V_{(1,1)A_1} \xrightarrow{Q^\dagger} 0\,, \tag{3.32}$$
$$0 \xrightarrow{Q^\dagger} V_{(1,1)A_2} \xrightarrow{Q^\dagger} 0\,.$$

Since the states in $V_{(1,1)A_1}$ and $V_{(1,1)A_2}$ are annihilated by both $Q$ and $Q^\dagger$, they are BPS *states* and remain unlifted under the exactly marginal deformation. In addition to these, there are also BPS states in $V_{(2)}$, which correspond one-to-one to the cohomology classes of cycle shape $(2)$.

The dimensions of the cohomology groups give the degeneracy of $\frac{1}{4}$-BPS states, which could be summarized into a BPS partition function, where the right-moving part of the states

in the $V_{(1,1)S_i}$, $V_{(1,1)A_i}$, and $V_{(2)}$ contribute $\overline{\chi_0^{\text{BPS}}}$, $2\overline{\chi_1^{\text{BPS}}}$, and $\overline{\chi_1^{\text{BPS}}}$ given in (2.49), respectively. We compare this result with the expression in (2.51), using the expansions of $\mathcal{S}_0$ and $\mathcal{S}_1$ in (B.3) and (B.5), and find an exact match up to $h = \frac{7}{2}$.

## 3.4  Large $N$ scaling

In our computation of the matrix elements of the supercharge $Q^{(\nu,\ell)}$, we adopted the convention where the normalization of states is defined in (2.5) and (2.6). In this convention, the matrix elements of $Q^{(\nu,\ell)}$ are independent of $N$, and thus all the $N$-dependence of the cohomologies arises from the stringy exclusion principle. In other words, there is a consistent $Q$-action on the holographic covering Hilbert space (3.6) that descends to the $Q$-action on the finite $N$ Hilbert spaces. However, to take the large $N$ limit, it is more appropriate to normalize the state on the left-hand side of (2.5). Before normalizing, a state with total length $\nu_0$ of non-trivial cycles has norm

$$\sqrt{\mathcal{N}_{i_a,w_a}} \sim N^{\frac{\nu_0}{2}} . \tag{3.33}$$

After normalization, the $N$-scaling of the supercharge $Q^{(\nu,\ell)}$ is given by the amount of change in the total length of non-trivial cycles:

$$Q^{(0,1)} \sim N^{\frac{1}{2}} , \quad Q^{(-1,0)} + Q^{(1,0)} \sim N^0 . \tag{3.34}$$

We see that $Q^{(0,1)}$ dominates in the large $N$ limit, leading naturally to the following conjecture:

**Conjecture 2** (Infinite $N$ supercharge cohomology). *For fixed charges, the supercharge $Q$-cohomology is isomorphic to the $Q^{(0,1)}$-cohomology in the infinite $N$ limit (without the stringy exclusion principle).*

This conjecture can be rephrased in the language of spectral sequence. Let us organize the Hilbert space of states (2.23) as a double complex $C^{m,n}$ graded by $(m,n) = (2\tilde{j} - \ell, \ell)$. The supercharge $Q^{0,1}$ has degree $(m,n) = (0,1)$, and the supercharge $Q^{(-1,0)} + Q^{(1,0)}$ has degree $(m,n) = (1,0)$. The zeroth page of the spectral seuqence is $E_0^{m,n} = C^{m,n}$ with the differential $d_0 = Q^{(0,1)}$. The first page is the $Q^{(0,1)}$ cohomology

$$E_1^{m,n} = H^m(C^{\bullet,n}, Q^{(0,1)}) , \tag{3.35}$$

with a differential $d_1$ induced by the supercharge $Q^{(-1,0)} + Q^{(1,0)}$. The infinite page $E_\infty^{m,n}$

gives the $Q$-cohomology

$$H^k\left(\text{Tot}(C), Q\right) = \bigoplus_{m+n=k} E_\infty^{m,n}, \quad \text{Tot}_k(C) := \bigoplus_{m+n=k} C^{m,n}. \tag{3.36}$$

Conjecture 2 implies that the spectral sequence is degenerate at the first page, i.e. $d_r$ vanish for $r \geq 1$ and $E_1^{m,n} \cong E_\infty^{m,n}$. The vanishing of $d_1$ implies that for any $Q^{(0,1)}$-closed state $|\Psi\rangle$, there exist a state $|\Psi'\rangle$ such that

$$(Q^{(-1,0)} + Q^{(1,0)})\,|\Psi\rangle = Q^{(0,1)}\,|\Psi'\rangle\,. \tag{3.37}$$

We checked this up to $h = \frac{5}{2}$ for states in $V_{(1,1)}$ and $V_{(2)}$.

It was conjectured in [7] that the supercharge cohomology classes in the infinite $N$ limit are dual to the non-interacting (multi-)supergraviton states on the bulk AdS vacuum. Since the multi-graviton states are given by products of single-graviton states and the number of gravitons is identified with the number of nontrivial cycles [27], we expect that products of single-cycle cohomology classes generate the full supercharge cohomology at infinite $N$. This is indeed the case by Conjecture 2, since the $Q^{(0,1)}$-action satisfies the Leibniz rule when acting on multi-cycle states. However, note that the $Q$-action does not satisfy the Leibniz rule, and hence, the $Q$-cohomology does not factorize into products of single-cycle cohomology classes. This means that the isomorphism map in Conjecture 2 is non-trivial, so that a $Q^{(0,1)}$-closed state that represents a $Q^{(0,1)}$-cohomology class might not be $Q$-closed.

The single-cycle $\frac{1}{2}$-BPS state and their $\text{SU}(1,1|2)$ descendants (generated by $L_{-1}$, $G^-_{-\frac{1}{2}}$, $G'^-_{-\frac{1}{2}}$, $K_0^{3-}$ actions) correspond to the $\frac{1}{4}$-BPS single-graviton states [27]. Since both the $\text{SU}(1,1|2)$ algebra and the $\frac{1}{2}$-BPS states are protected under the deformation, we expect these single-cycle states to remain $\frac{1}{4}$-BPS. We have constructed the single-cycle $Q^{(0,1)}$-cohomology classes in the infinite $N$ limit up to $h = \frac{5}{2}$ and found agreement with the counting of the above single-cycle $\frac{1}{4}$-BPS states. This result provides evidence supporting Conjecture 2 and the conjecture in [7].

# 4 Monotone and fortuitous cohomologies

## 4.1 Classification

A general classification of supercharge $Q$-cohomology classes was proposed in [7] based on their properties at large $N$. The classification relies on a property that the Hilbert space (as a vector space) of the finite $N$ theory can be expressed as a quotient of the Hilbert

space of the infinite $N$ theory by certain equivalent relations. A monotone cohomology class has a representative that can be pulled back to a representative of a nontrivial cohomology class in the infinite $N$ theory. By contrast, the pullback of a representative of a fortuitous cohomology class is not $Q$-closed.

Let us see explicitly how this classification applies to the D1-D5 CFT. The supercharge $Q$-action under the exactly marginal deformation at first order, as discussed in Section 3.2, is independent of the total length of the cycles and can be lifted to the infinite $N$ limit. By relating the cohomology classes at finite $N$ and infinite $N$, we have the following definitions for the monotone and fortuitous cohomology classes:

**Definition 1** (Monotone cohomology). *A cohomology class is monotone if it admits a representative that also represents a non-trivial cohomology class in the infinite $N$ limit.*

**Definition 2** (Fortuitous cohomology). *A cohomology class is fortuitous if the $Q$-action on any representative results in a state whose total length of non-trivial cycles exceeds $N$.*[9]

The above definitions are equivalent to the original definitions in [7], where monotone cohomology classes can be lifted to cohomology classes in the infinite $N$ limit, while fortuitous cohomology classes cannot.

## 4.2   Monotone cohomology and partition function

At finite $N$, the monotone cohomology is given by imposing the stringy exclusion principle on the infinite $N$ cohomology, which by Conjecture 2 is isomorphic to the $Q^{(0,1)}$-cohomology. This relation allows us to write down a partition function that counts the number of monotone cohomology classes.

Let us start by computing the partition function for the single-cycle $Q^{(0,1)}$-cohomology. As argued in Section 3.4, they are represented by the left-moving $SU(1,1|2)$ descendants of the $\frac{1}{2}$-BPS states (2.21). The $SU(1,1|2)$ descendants of a primary state are counted by the $SU(1,1|2)$ character. In the NS sector and weighted by $(-1)^F$, it is given by

$$\hat{\chi}_j = \frac{q^{j-\frac{k}{4}}}{(1-q)(y-y^{-1})} \left[ (y^{2j+1} - y^{-2j-1}) - 2q^{\frac{1}{2}}(y^{2j} - y^{-2j}) + q(y^{2j-1} - y^{-(2j-1)}) \right] . \quad (4.1)$$

Summing over the characters for all the single-cycle $\frac{1}{2}$-BPS states in the Clifford module

---

[9]Let $|\Phi\rangle$ be a fortuitous representative, then $\langle\Phi| Q |\{\Phi\}\rangle = 0$ for any $|\Psi\rangle$ with a total length of nontrivial cycles at most $N$.

(2.21), we find the partition function for the single-cycle $Q^{(0,1)}$-cohomology:

$$z^{Q^{(0,1)}}(p,q,y,\bar{q},\bar{y}) = \sum_{k=1}^{\infty} p^k \left( \hat{\chi}_{\frac{k+1}{2}}(q,y) - 2\hat{\chi}_{\frac{k}{2}}(q,y) + \hat{\chi}_{\frac{k-1}{2}}(q,y) \right) \bar{q}^{\frac{k-2}{4}} \bar{y}^{k-1} \left( 1 - \bar{q}^{\frac{1}{2}}\bar{y} \right)^2 ,$$

(4.2)

where we introduce a fugacity $p$ for the length of the cycles.

The partition function for the (multi-cycle) $Q^{(0,1)}$-cohomology is given by the plethystic exponential

$$\mathcal{Z}^{Q^{(0,1)}}(p,q,\bar{q},y,\bar{y}) = \exp\left[ \sum_{r=1}^{\infty} \frac{1}{r} z^{Q^{(0,1)}}(p^r,q^r,y^r,\bar{q}^r,\bar{y}^r) \right] .$$

(4.3)

Imposing the stringy exclusion principle fixes the total length (of the cycles) to equal $N$, and amounts to extracting the coefficient of $p^N$,

$$Z_N^{Q^{(0,1)}} = \mathcal{Z}^{Q^{(0,1)}}(p,q,y,\bar{q},\bar{y})\Big|_{p^N} ,$$

(4.4)

which, by Conjecture 2, is equal to the monotone partition function,

$$Z_N^{\text{mon}} = Z_{(N)}^{Q^{(0,1)}} .$$

(4.5)

The modified index for monotone states can be computed by

$$I_N^{\text{mon}} = \frac{1}{2} \bar{q}^{\frac{N}{4}-1} \partial_{\bar{y}}^2 Z_N^{\text{grav}} .$$

(4.6)

For instance, the expansions of the $N=2$ and 3 monotone indices are

$$I_{N=2}^{\text{mon}} = \frac{2}{q^{\frac{1}{2}}} + \left( y + \frac{1}{y} \right) - q^{\frac{1}{2}} \left( 6y^2 + 4 + \frac{6}{y^2} \right) + q \left( y^3 + 6y + \frac{6}{y} + \frac{1}{y^3} \right) + O\left( q^{\frac{3}{2}} \right) ,$$

$$I_{N=3}^{\text{mon}} = \frac{3}{q^{\frac{3}{4}}} + q^{\frac{1}{4}} \left( y^2 + 1 + \frac{1}{y^2} \right) - 4q^{\frac{3}{4}} \left( 2y^3 + y + \frac{1}{y} + \frac{2}{y^3} \right) + O\left( q^{\frac{5}{4}} \right) .$$

(4.7)

We expand the $N=2$ and 3 monotone indices up to $q^3$ in (B.10) and (B.11) in Appendix B.

Taking the difference between the full index and the monotone index, we obtain the fortuitous index,

$$I_N^{\text{for}} = I_{\text{Sym}^N(T^4),\text{NS}} - I_N^{\text{mon}} .$$

(4.8)

For instance, the expansions of the $N = 2$ and $3$ fortuitous indices are

$$
\begin{aligned}
I_{N=2}^{\text{for}} &= -8\sqrt{q} + q\left(33y + \frac{33}{y}\right) + q^{\frac{3}{2}}\left(-52y^2 - 194 - \frac{52}{y^2}\right) + O\left(q^{\frac{7}{2}}\right) \\
I_{N=3}^{\text{for}} &= 7\sqrt[4]{q} + q^{\frac{3}{4}}\left(-52y - \frac{52}{y}\right) + q^{\frac{5}{4}}\left(146y^2 + 506 + \frac{146}{y^2}\right) + O\left(q^{\frac{13}{4}}\right)
\end{aligned}
\tag{4.9}
$$

We give the expansions of the $N = 2$ and $3$ fortuitous indices up to $q^3$ in (B.12) and (B.13) in Appendix B.

## 4.3 Fortuitous partition function at $N = 2$

To compare the monotone partition function with the full BPS partition function at $N = 2$ given by (2.51), we consider the expansion

$$
Z_{\text{mon}}^{(N=2)} = \mathcal{S}_0^{\text{mon}}\overline{\chi_0^{\text{BPS}}} + \mathcal{S}_1^{\text{mon}}\overline{\chi_1^{\text{BPS}}},
\tag{4.10}
$$

where $\overline{\chi_0^{\text{BPS}}}$ and $\overline{\chi_1^{\text{BPS}}}$ are defined in (2.49), and the expansions for $\mathcal{S}_0^{\text{mon}}$ and $\mathcal{S}_1^{\text{mon}}$ are

$$
\begin{aligned}
\mathcal{S}_0^{\text{mon}} &= \frac{1}{q^{\frac{1}{2}}} - 2\left(y + \frac{1}{y}\right) + q^{\frac{1}{2}}\left(2y^2 + \frac{2}{y^2} + 9\right) + O(q), \\
\mathcal{S}_1^{\text{mon}} &= -5\left(y + \frac{1}{y}\right) + q^{\frac{1}{2}}\left(10y^2 + 22 + \frac{10}{y^2}\right) + O(q).
\end{aligned}
\tag{4.11}
$$

We give the expansions of $\mathcal{S}_0^{\text{mon}}$ and $\mathcal{S}_1^{\text{mon}}$ up to $q^3$ in (4.11) and (B.7) in Appendix B. Taking the difference between the BPS partition function $Z_{N=2}^{\text{BPS}}$ in (2.51) and the monotone partition function $Z_{N=2}^{\text{mon}}$, we find the fortuitous partition function

$$
\begin{aligned}
Z_{N=2}^{\text{for}} &= Z_{N=2}^{\text{BPS}} - Z_{N=2}^{\text{mon}} = (\mathcal{S}_0 - \mathcal{S}_0^{\text{mon}})\overline{\chi_0^{\text{BPS}}} + (\mathcal{S}_1 - \mathcal{S}_1^{\text{mon}})\overline{\chi_1^{\text{BPS}}} \\
&= \mathcal{S}_0^{\text{for}}\overline{\chi_0^{\text{BPS}}} + \mathcal{S}_1^{\text{for}}\overline{\chi_1^{\text{BPS}}}.
\end{aligned}
\tag{4.12}
$$

The expansions for $\mathcal{S}_0^{\text{for}}$ and $\mathcal{S}_1^{\text{for}}$ are

$$
\begin{aligned}
\mathcal{S}_0^{\text{for}} &= -2(y^{-1} + y)q + \left(22 + \frac{8}{y^2} + 8y^2\right)q^{3/2} + +O\left(q^2\right), \\
\mathcal{S}_1^{\text{for}} &= 8q^{\frac{1}{2}} - q\left(37y + \frac{37}{y}\right) + q^{\frac{3}{2}}\left(68y^2 + 238 + \frac{68}{y^2}\right) + O\left(q^2\right).
\end{aligned}
\tag{4.13}
$$

We give the expansions of $\mathcal{S}_0^{\text{for}}$ and $\mathcal{S}_1^{\text{for}}$ up to $q^3$ in (B.8) and (B.9) in Appendix B. Our results indicate that, for large conformal dimension $h$, the number of fortuitous states dominates

over that of the monotone states. To see this more explicitly, let us define degeneracies as

$$d_0(h) = \mathrm{abs}\left(\mathcal{S}_0\big|_{q^{h-\frac{1}{2}},y=1}\right) , \quad d_1(h) = \mathrm{abs}\left(\mathcal{S}_1\big|_{q^{h-\frac{1}{2}},y=1}\right) , \tag{4.14}$$

and plot them in Figure 1.

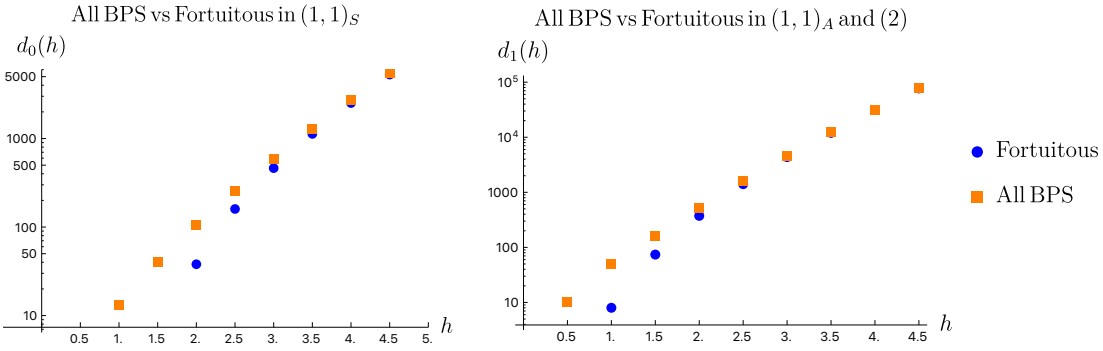

Figure 1: Comparison of $d_i(h)$ between $\mathcal{S}_i$ and $\mathcal{S}_i^{\mathrm{for}}$.

Although the *counting* of $Q^{(0,1)}$-cohomology classes with the stringy exclusion principle imposed gives us the monotone partition function by Conjecture 2, we emphasize that the $Q^{(0,1)}$-cohomology classes and (monotone) $Q$-cohomology classes are not the same with and without imposing the stringy exclusion principle. As was already pointed out in Section 3.4, since the supercharge $Q$ at finite $N$ does not satisfy the Leibniz rule, the multi-cycle monotone cohomology classes are not given by the products of the single-cycle ones.

It was conjectured in [7] that, under the AdS/CFT correspondence, the monotone cohomology classes are dual to bulk states arising from quantizing the moduli space of smooth horizonless solutions to the supergravity equations. The quantization of the $\frac{1}{2}$-BPS Lunin-Mathur geometries [15–18] and the $\frac{1}{4}$-BPS superstrata geometries [19–26] were carried out in [51–54], and their counting agrees with with the monotone partition function (4.5).[10]

## 4.4 Explicit fortuitous cohomology representatives at $N = 2$

Let us examine the explicit form of fortuitous states in the case of $N = 2$. We first consider all possible actions of the first-order deformed supercharge $Q$ that map to or from states in $V_{(2)}$ or $V_{(1,1)}$ without imposing the stringy exclusion principle, and then cross out the terms

---

[10]The Lunin-Mathur and superstrata geometries are often labeled as $\frac{1}{4}$-BPS and $\frac{1}{8}$-BPS in the literature, instead of $\frac{1}{2}$-BPS and $\frac{1}{4}$-BPS as in this paper. The difference in convention arises from whether the AdS vacuum $\mathrm{AdS}_3 \times \mathrm{S}^3 \times T^4$ with 16 supercharges is considered "1-BPS" or $\frac{1}{2}$-BPS relative to the 32 supercharges of type II in flat space.

with cycle shapes (3) or $(1,2)$ which are eliminated by the $N=2$ stringy exclusion principle:

$$0 \xrightarrow{\;Q\;} V_{(1,1)S_1} \xrightarrow{\;Q^{(\pm 1,0)}\;} V_{(2)} \xrightarrow{\;Q^{(\pm 1,0)}\;} V_{(1,1)S_2} \xrightarrow{\;Q^{(\pm 1,0)}\;} 0$$

$$\searrow_{Q^{(0,1)}} \quad \searrow_{Q^{(0,1)}} \quad \searrow_{Q^{(0,1)}}$$

$$\cancel{V_{(1,2)}} \qquad \cancel{V_{(3)}} \qquad \cancel{V_{(1,2)}} \tag{4.15}$$

and

$$0 \xrightarrow{\;Q\;} V_{(1,1)A_i} \xrightarrow{\;Q^{(\pm 1,0)}\;} 0$$

$$\searrow_{Q^{(0,1)}}$$

$$\cancel{V_{(1,2)}} \tag{4.16}$$

where the spaces $V_{(1,1)\bullet}$ were defined in (3.24), (3.25), (3.26) and (3.27). Note that the complexes (4.15) and (4.16) reduce to complexes (3.31) after removing the crossed-out terms.

The fortuitous cohomology classes are represented by the states that are mapped by $Q$ to states in $V_{(3)}$ or $V_{(1,2)}$. Let us start by looking at the cases with the lowest dimension $h$ in each sector. The representatives in $V_{(2)}$ and $V_{(1,1)A_i}$ of the first fortuitous cohomology classes appear at $h=1, j=0$, whereas those in $V_{(1,1)S_i}$ appear at $h=\frac{3}{2}, j=\frac{1}{2}$.

$\underline{h=1, \; j=0}$

(2) **sector**    The representatives are

$$(\psi_0^- \bar\alpha_{-\frac{1}{2}}^2 + \bar\psi_0^- \alpha_{-\frac{1}{2}}^2) |2_{\pm\pm}\rangle , \quad \bar\psi_0^- \alpha_{-\frac{1}{2}}^1 |2_{\pm\pm}\rangle . \tag{4.17}$$

$(1,1)_{A_i}$ **sector**    The representatives are the two quartets in (3.26) and (3.27) with the state $|\Psi_{[ij]}\rangle$ given by[11]

$$(\psi_{-\frac{1}{2}}^{(1)-} \psi_{-\frac{1}{2}}^{(1)+} - \psi_{-\frac{1}{2}}^{(2)-} \psi_{-\frac{1}{2}}^{(2)+})|1_{--,--}, 1_{--.--}\rangle,$$

$$(\psi_{-\frac{1}{2}}^{(1)-} \bar\psi_{-\frac{1}{2}}^{(1)+} - \psi_{-\frac{1}{2}}^{(2)-} \bar\psi_{-\frac{1}{2}}^{(2)+})|1_{--,--}, 1_{--.--}\rangle, \tag{4.18}$$

$$(\bar\psi_{-\frac{1}{2}}^{(1)-} \bar\psi_{-\frac{1}{2}}^{(1)+} - \bar\psi_{-\frac{1}{2}}^{(2)-} \bar\psi_{-\frac{1}{2}}^{(2)+})|1_{--,--}, 1_{--.--}\rangle .$$

The two states (4.17) in $V_{(2)}$ and the three states (4.18) in each of $V_{(1,1)A_1}$ and $V_{(1,1)A_2}$ account for the term $(2+3\times 2)q^{\frac{1}{2}}$ in $\mathcal{S}_1^{\text{for}}$ of the fortuitous partition function (4.12) (with the expansion (4.13)).

---

[11]The superscripts (1) and (2) indicate the generators acting on the first copy and the second copy, respectively.

$\underline{h = \frac{3}{2}, \ j = \frac{1}{2}}$

**(2) sector** There are nine fortuitous cohomology classes with the explicit representatives:

$$\bar{\psi}^+_{-\frac{1}{2}}\psi^-_0\psi^+_{-\frac{1}{2}}\bar{\psi}^-_0\,|2_{\pm\pm}\rangle\,,\quad(-\psi^-_0\psi^+_{-\frac{1}{2}}\bar{\alpha}^2_{-\frac{1}{2}}+\psi^+_{-\frac{1}{2}}\bar{\psi}^-_0\alpha^2_{-\frac{1}{2}})\,|2_{\pm\pm}\rangle\,,\quad\psi^+_{-\frac{1}{2}}\bar{\psi}^-_0\alpha^1_{-\frac{1}{2}}\,|2_{\pm\pm}\rangle$$

$$\bar{\psi}^+_{-\frac{1}{2}}\bar{\psi}^-_0\alpha^1_{-\frac{1}{2}}\,|2_{\pm\pm}\rangle\,,\quad(\bar{\psi}^+_{-\frac{1}{2}}\psi^-_0\bar{\alpha}^2_{-\frac{1}{2}}+\bar{\psi}^+_{-\frac{1}{2}}\bar{\psi}^-_0\alpha^2_{-\frac{1}{2}})\,|2_{\pm\pm}\rangle\quad(\psi^+_{-\frac{1}{2}}\bar{\psi}^-_{-\frac{1}{2}})\,|2_{\pm\pm}\rangle\,,\qquad(4.19)$$

$$\bar{\psi}^+_{-\frac{1}{2}}\bar{\psi}^-_{-\frac{1}{2}}\,|2_{\pm\pm}\rangle\,,\quad(2\alpha^1_{-\frac{1}{2}}\bar{\alpha}^+_{-\frac{1}{2}}+\bar{\psi}^+_{-1}\psi^-_0)\,|2_{\pm\pm}\rangle\,,\qquad\psi^-_{-\frac{1}{2}}\psi^+_{-\frac{1}{2}}\,|2_{\pm\pm}\rangle\,.$$

Together with 14 fortuitous cohomology classes in each of the $(1,1)_{A_1}$ and $(1,1)_{A_2}$ sectors, they correspond to the term $-q(37y+37y^{-1})$ in $\mathcal{S}^{\text{for}}_1$ (4.13).

$(1,1)_S$ **sector** The representatives of the former cohomology classes are the two quartets in (3.24) and (3.25) with the state $\Psi_{[ij]}$ given by

$$\begin{aligned}(\bar{\psi}^{(1)+}_{-\frac{1}{2}}\psi^{(1)-}_{-\frac{1}{2}}\psi^{(1)+}_{-\frac{1}{2}}&+\bar{\psi}^{(2)+}_{-\frac{1}{2}}\psi^{(1)-}_{-\frac{1}{2}}\psi^{(1)+}_{-\frac{1}{2}}+\psi^{(2)+}_{-\frac{1}{2}}\psi^{(1)+}_{-\frac{1}{2}}\bar{\psi}^{(1)-}_{-\frac{1}{2}}\\&+\bar{\psi}^{(2)+}_{-\frac{1}{2}}\psi^{(2)-}_{-\frac{1}{2}}\psi^{(2)+}_{-\frac{1}{2}}+\bar{\psi}^{(1)+}_{-\frac{1}{2}}\psi^{(2)-}_{-\frac{1}{2}}\psi^{(2)+}_{-\frac{1}{2}}+\psi^{(1)+}_{-\frac{1}{2}}\psi^{(2)+}_{-\frac{1}{2}}\bar{\psi}^{(2)-}_{-\frac{1}{2}})|1_{--,--},1_{--.--}\rangle\,,\\(-\bar{\psi}^{(1)+}_{-\frac{1}{2}}\psi^{(1)+}_{-\frac{1}{2}}\bar{\psi}^{(1)-}_{-\frac{1}{2}}&-\bar{\psi}^{(2)+}_{-\frac{1}{2}}\bar{\psi}^{(1)+}_{-\frac{1}{2}}\psi^{(1)-}_{-\frac{1}{2}}+\psi^{(2)+}_{-\frac{1}{2}}\bar{\psi}^{(1)+}_{-\frac{1}{2}}\bar{\psi}^{(1)-}_{-\frac{1}{2}}\\&-\bar{\psi}^{(2)+}_{-\frac{1}{2}}\psi^{(2)+}_{-\frac{1}{2}}\bar{\psi}^{(2)-}_{-\frac{1}{2}}-\bar{\psi}^{(1)+}_{-\frac{1}{2}}\bar{\psi}^{(2)+}_{-\frac{1}{2}}\psi^{(2)-}_{-\frac{1}{2}}+\psi^{(1)+}_{-\frac{1}{2}}\bar{\psi}^{(2)+}_{-\frac{1}{2}}\bar{\psi}^{(2)-}_{-\frac{1}{2}})|1_{--,--},1_{--.--}\rangle\,.\end{aligned}\qquad(4.20)$$

The latter cohomology classes have similar representatives. Together, they correspond to the term $-2(y+y^{-1})q$ in $\mathcal{S}^{\text{for}}_0$ in the fortuitous partition function (4.12) (with the expansion (4.13)).

We carry out the computation up to $h=\frac{5}{2}$ and find exact agreement with the fortuitous partition function $Z^{\text{for}}_{N=2}$ in (4.12) (with the expansion in (B.8) and and (B.9)).

# 5 Composite BPS states

This section uses single-cycle BPS states to construct composite BPS states. We depart from the $Q$-cohomology framework and consider actual BPS *states* because it is simpler analyzing the $Q^\dagger$ action than establishing non-$Q$-exactness.

## 5.1 When is a two-cycle state BPS?

Let us consider a two-cycle state in $\mathrm{Sym}^N(T^4)$[12]

$$|\{(w_1)_{i,\pm\pm}, (w_2)_{j,\pm\pm}\}\rangle. \tag{5.1}$$

As we have seen in Section 3.3, the right-moving Clifford algebra imposes selection rules on the $Q$-action, which could forbid certain joining and splitting processes. Let us decompose the product state (5.1) into four quartets under the diagonal right-moving Clifford algebra:

$$V_{(w_1,w_2)P_1} := \mathrm{span}
\begin{bmatrix}
(\tilde{\psi}^{(1)+}_{-\frac{1}{2}} + \tilde{\psi}^{(2)+}_{-\frac{1}{2}})(\tilde{\bar{\psi}}^{(1)+}_{-\frac{1}{2}} + \tilde{\bar{\psi}}^{(2)+}_{-\frac{1}{2}})\,|\Psi_{ij}\rangle \\
(\tilde{\bar{\psi}}^{(1)+}_{-\frac{1}{2}} + \tilde{\bar{\psi}}^{(2)+}_{-\frac{1}{2}})\,|\Psi_{ij}\rangle \\
(\tilde{\psi}^{(1)+}_{-\frac{1}{2}} + \tilde{\psi}^{(2)+}_{-\frac{1}{2}})\,|\Psi_{ij}\rangle \\
|\Psi_{ij}\rangle
\end{bmatrix}
,\quad
\tilde{j} =
\begin{bmatrix}
\frac{w_1+w_2}{2} \\
\frac{w_1+w_2-1}{2} \\
\frac{w_1+w_2-1}{2} \\
\frac{w_1+w_2-2}{2}
\end{bmatrix}
, \tag{5.2}$$

$$V_{(w_1,w_2)P_2} := \mathrm{span}
\begin{bmatrix}
\tilde{\psi}^{(1)+}_{-\frac{1}{2}}\tilde{\psi}^{(2)+}_{-\frac{1}{2}}\tilde{\bar{\psi}}^{(1)+}_{-\frac{1}{2}}\tilde{\bar{\psi}}^{(2)+}_{-\frac{1}{2}}\,|\Psi_{ij}\rangle \\
\tilde{\psi}^{(1)+}_{-\frac{1}{2}}\tilde{\psi}^{(2)+}_{-\frac{1}{2}}(\tilde{\bar{\psi}}^{(1)+}_{-\frac{1}{2}} - \tilde{\bar{\psi}}^{(2)+}_{-\frac{1}{2}})\,|\Psi_{ij}\rangle \\
(\tilde{\psi}^{(1)+}_{-\frac{1}{2}} - \tilde{\psi}^{(2)+}_{-\frac{1}{2}})\tilde{\bar{\psi}}^{(1)+}_{-\frac{1}{2}}\tilde{\bar{\psi}}^{(2)+}_{-\frac{1}{2}}\,|\Psi_{ij}\rangle \\
(\tilde{\psi}^{(1)+}_{-\frac{1}{2}} - \tilde{\psi}^{(2)+}_{-\frac{1}{2}})(\tilde{\bar{\psi}}^{(1)+}_{-\frac{1}{2}} - \tilde{\bar{\psi}}^{(2)+}_{-\frac{1}{2}})\,|\Psi_{ij}\rangle
\end{bmatrix}
,\quad
\tilde{j} =
\begin{bmatrix}
\frac{w_1+w_2+2}{2} \\
\frac{w_1+w_2+1}{2} \\
\frac{w_1+w_2+1}{2} \\
\frac{w_1+w_2}{2}
\end{bmatrix}
, \tag{5.3}$$

$$V_{(w_1,w_2)P_3} := \mathrm{span}
\begin{bmatrix}
\tilde{\psi}^{(1)+}_{-\frac{1}{2}}\tilde{\psi}^{(2)+}_{-\frac{1}{2}}(\tilde{\bar{\psi}}^{(1)+}_{-\frac{1}{2}} + \tilde{\bar{\psi}}^{(2)+}_{-\frac{1}{2}})\,|\Psi_{ij}\rangle \\
(\tilde{\psi}^{(1)+}_{-\frac{1}{2}} - \tilde{\psi}^{(2)+}_{-\frac{1}{2}})(\tilde{\bar{\psi}}^{(1)+}_{-\frac{1}{2}} + \tilde{\bar{\psi}}^{(2)+}_{-\frac{1}{2}})\,|\Psi_{ij}\rangle \\
\tilde{\psi}^{(1)+}_{-\frac{1}{2}}\tilde{\psi}^{(2)+}_{-\frac{1}{2}}\,|\Psi_{ij}\rangle \\
(\tilde{\psi}^{(1)+}_{-\frac{1}{2}} - \tilde{\psi}^{(2)+}_{-\frac{1}{2}})\,|\Psi_{ij}\rangle
\end{bmatrix}
,\quad
\tilde{j} =
\begin{bmatrix}
\frac{w_1+w_2+1}{2} \\
\frac{w_1+w_2}{2} \\
\frac{w_1+w_2}{2} \\
\frac{w_1+w_2-1}{2}
\end{bmatrix}
, \tag{5.4}$$

$$V_{(w_1,w_2)P_4} := \mathrm{span}
\begin{bmatrix}
\tilde{\bar{\psi}}^{(1)+}_{-\frac{1}{2}}\tilde{\bar{\psi}}^{(2)+}_{-\frac{1}{2}}(\tilde{\psi}^{(1)+}_{-\frac{1}{2}} + \tilde{\psi}^{(2)+}_{-\frac{1}{2}})\,|\Psi_{ij}\rangle \\
(\tilde{\bar{\psi}}^{(1)+}_{-\frac{1}{2}} - \tilde{\bar{\psi}}^{(2)+}_{-\frac{1}{2}})(\tilde{\psi}^{(1)+}_{-\frac{1}{2}} + \tilde{\psi}^{(2)+}_{-\frac{1}{2}})\,|\Psi_{ij}\rangle \\
\tilde{\bar{\psi}}^{(1)+}_{-\frac{1}{2}}\tilde{\bar{\psi}}^{(2)+}_{-\frac{1}{2}}\,|\Psi_{ij}\rangle \\
(\tilde{\bar{\psi}}^{(1)+}_{-\frac{1}{2}} - \tilde{\bar{\psi}}^{(2)+}_{-\frac{1}{2}})\,|\Psi_{ij}\rangle
\end{bmatrix}
,\quad
\tilde{j} =
\begin{bmatrix}
\frac{w_1+w_2+1}{2} \\
\frac{w_1+w_2}{2} \\
\frac{w_1+w_2}{2} \\
\frac{w_1+w_2-1}{2}
\end{bmatrix}
. \tag{5.5}$$

---

[12]At this point we do not impose any further assumptions (e.g. closedness).

where $|\Psi_{ij}\rangle := |(w_1)_{i,--}, (w_2)_{j,--}\rangle$.[13]

In the remainder, we only consider the states in $V_{(w_1,w_2)P_{3,4}}$. Adapting the arguments of Section 3.3, the possible $Q$ and $Q^\dagger$-actions on the states in $V_{(w_1,w_2)P_{3,4}}$ are the following:

$$
V_{(w_1,w_2)P_{3,4}} \quad
\begin{array}{l}
\xrightarrow{\;Q^{(1,0)}\;} \quad \bigoplus_m V_{(w_1,w_2-n,n)} \oplus V_{(w_1-n,w_2,n)} \\[2mm]
\xrightarrow{\;Q^{(0,1)}\;} \quad V_{(w_1,w_2+1)P_{3,4}} \oplus V_{(w_1+1,w_2)P_{3,4}} \\[2mm]
\xrightarrow{\;Q^{(-1,0)}\;} \quad \cancel{V_{(w_1+w_2)}}
\end{array}
\tag{5.6}
$$

and

$$
\begin{array}{l}
\cancel{V_{(w_1+w_2)}} \quad\xleftarrow{\;(Q^\dagger)^{(-1,0)}\;} \\[2mm]
V_{(w_1,w_2-1)P_{3,4}} \oplus V_{(w_1-1,w_2)P_{3,4}} \quad\xleftarrow{\;(Q^\dagger)^{(0,-1)}\;}\quad V_{(w_1,w_2)P_{3,4}} \\[2mm]
\bigoplus_n V_{(w_1,w_2-n,n)} \oplus V_{(w_1-n,w_2,n)} \quad\xleftarrow{\;(Q^\dagger)^{(1,0)}\;}
\end{array}
\tag{5.7}
$$

where $\cancel{V_{(w_1+w_2)}}$ is forbidden by the R-charge conservation. To see this, simply observe that the quartets in $V_{(w_1+w_2)}$ have R-charges

$$
\tilde{j} = \begin{bmatrix} \frac{w_1+w_2+1}{2} \\[2mm] \frac{w_1+w_2}{2} \\[2mm] \frac{w_1+w_2}{2} \\[2mm] \frac{w_1+w_2-1}{2} \end{bmatrix},
\tag{5.8}
$$

identical to those in $V_{(w_1,w_2)P_{3,4}}$. Also note that R-charge conservation does allow states in $V_{(w_1+w_2)}$ to map to states in $V_{(w_1,w_2)P_1}$ or map from states in $V_{(w_1,w_2)P_2}$.

Our goal is to find BPS states in $V_{(w_1,w_2)P_{3,4}}$. A first natural ansatz is to consider the *composite* two-cycle state

$$
u^i v^j \,|\{(w_1)_i, (w_2)_j\}\rangle \in V_{(w_1,w_2)P_{3,4}},
\tag{5.9}
$$

where we suppressed the anti-holomorphic indices $\alpha, \beta$, since they have already been specified by the fact that the state is inside $V_{(w_1,w_2)P_{3,4}}$. Since $Q^{(-1,0)}$ annihilates this state, the only

---

[13]More precisely, we map the states in (5.2) to $S_N$ invariants by summing over their $S_N$ images as in (2.5).

nontrivial $Q$-actions are

$$
Q^{(1,0)}u^i v^j \left|\{(w_1)_i, (w_2)_j\}\right\rangle = \sum_{n=1}^{w_1-1} u^i v^j c^{(1,0)}_{w_1-n,n;i,k_1,k_2} \left|\{(w_1-n)_{k_1}, n_{k_2}, (w_2)_j\}\right\rangle
$$
$$
+ \sum_{n=1}^{w_2-1} u^i v^j c^{(1,0)}_{w_2-n,n;j,k_1,k_2} \left|\{(w_1)_i, (w_2-n)_{k_1}, n_{k_2}\}\right\rangle , \qquad (5.10)
$$
$$
Q^{(0,1)}u^i v^j \left|\{(w_1)_i, (w_2)_j\}\right\rangle = u^i v^j c^{(0,1)}_{w;i,k} \left|\{(w_1)_i, (w_2+1)_k\}\right\rangle
$$
$$
+ u^i v^j c^{(0,1)}_{w;i,k} \left|\{(w_1+1)_k, (w_2)_j\}\right\rangle ,
$$

where the repeated indices are summed. The state (5.9) is $Q$-closed, if we choose the vectors $u^i$ and $v^j$ that satisfy

$$
u^i c^{(1,0)}_{w_1-n,n;i,k_1 k_2} = 0 , \quad u^i c^{(0,1)}_{w;i,k} = 0 , \quad v^j c^{(1,0)}_{w_2-n,n;j,k_1 k_2} = 0 , \quad v^j c^{(0,1)}_{w;j,k} = 0 . \qquad (5.11)
$$

These conditions are the same conditions for the states

$$
u^i \left|\{(w_1)_{i,\pm\pm}\}\right\rangle , \quad v^j \left|\{(w_2)_{j,\pm\pm}\}\right\rangle \qquad (5.12)
$$

being $Q$-closed. A quick way to see that the $Q$-closedness of the single-cycle states (5.12) implies the $Q$-closedness of the composite two-cycle state (5.9) is to note that the supercharge $Q^{(1,0)} + Q^{(0,1)}$ satisfies the Leibniz rule when acting on multi-cycle states.

By a similar argument, we know that if the single-cycle states (5.12) are $Q^\dagger$-closed then the composite two-cycle state (5.9) is also $Q^\dagger$-closed.

Therefore, to guarantee that the state (5.9) is BPS, we will assume that the states in (5.12) are BPS. Note that the requirement of a single-cycle BPS *state* is stronger than a single-cycle BPS *cohomology*; given the latter, the corresponding BPS state under Hodge duality can generally involve the addition of a multi-cycle $Q$-exact term. One may hence worry that our premise is too strong to be useful. Fortunately, there are plenty of examples of single-cycle BPS states. As discussed, the left-moving $\mathrm{SU}(1,1|2)$ descendants of the single-cycle $\frac{1}{2}$-BPS states are monotone BPS states. We have also found explicit examples of length-2 single-cycle fortuitous BPS states in Section 3.3. We believe that there are more single-cycle fortuitous BPS states with longer lengths.

Let us discuss when the composite two-cycle state is monotone and when it is fortuitous. This distinction only concerns the $Q$ action and not $Q^\dagger$.[14] In the following discussion, it is important to note that the state $u^i \left|\{(w_1)_{i,\pm\pm}\}\right\rangle$ makes sense as a family of states, one in

---

[14]In the full theory, $Q$ and $Q^\dagger$ are on equal footing, but we have chosen to study the $\frac{1}{4}$-BPS sector satisfying $\tilde{h} = \tilde{j}$ instead of $\tilde{h} = -\tilde{j}$ and this choice underlies the dichotomy between $Q$ and $Q^\dagger$ throughout our analysis.

each of the $\mathrm{Sym}^N(T^4)$ theory for $N \geq w_1$, and similarly for the states $v^j \, |\{(w_2)_{j,\pm\pm}\}\rangle$ and $u^i v^j \, |\{(w_1)_i, (w_2)_j\}\rangle$. As commented at the beginning of Section 3.4, we have normalized our states so that the $Q$-action on finite-$N$ Hilbert spaces consistently descends from a $Q$-action on the holographic covering Hilbert space through quotients by the stringy exclusion principle.

If the states in (5.12) are both monotone, then the state (5.9) is also monotone. Suppose the state $u^i \, |\{(w_1)_{i,\pm\pm}\}\rangle$ is fortuitous when $N = w_1$, i.e. when $N > w_1$, we have

$$Q u^i \, |\{(w_1)_{i,\pm\pm}\}\rangle \in V_{(w_1+1)} \, . \tag{5.13}$$

Then we must have

$$Q u^i v^j \, |\{(w_1)_i, (w_2)_j\}\rangle \in V_{(w_1+1,w_2)} \oplus V_{(w_1,w_2+1)}, \tag{5.14}$$

where $N > w_1 + w_2$. Therefore, we know that the composite state $u^i v^j \, |\{(w_1)_i, (w_2)_j\}\rangle$ is fortuitous when $N = w_1 + w_2$.

Our construction above does not obviously generalize to more than two constituents. Let $(w_1, w_2, w_3)_P$ be the projection to one of the 16 quartets, where the possible R-charges are

$$\tilde{j} = \begin{bmatrix} \frac{\ell+n+1}{2} \\ \frac{\ell+n}{2} \\ \frac{\ell+n}{2} \\ \frac{\ell+n-1}{2} \end{bmatrix} \qquad \text{with } \ell = w_1 + w_2 + w_3 \text{ and } -2 \leq n \leq 2. \tag{5.15}$$

Consider the maps

$$(w_1 + w_2, w_3) \oplus (2 \text{ perm}) \quad \xrightarrow{\;Q^{(1,0)}\;} \quad (w_1, w_2, w_3)_P \tag{5.16}$$

and

$$(w_1, w_2, w_3)_P \quad \xrightarrow{\;Q^{(-1,0)}\;} \quad (w_1 + w_2, w_3) \bigoplus (2 \text{ perm}) \tag{5.17}$$

There are 4 quartets of the form $(w_1 + w_2, w_3)$ with R-charges,

$$\tilde{j} = \begin{bmatrix} \frac{\ell+2}{2} \\ \frac{\ell+1}{2} \\ \frac{\ell+1}{2} \\ \frac{\ell}{2} \end{bmatrix}, \quad \begin{bmatrix} \frac{\ell+1}{2} \\ \frac{\ell}{2} \\ \frac{\ell}{2} \\ \frac{\ell-1}{2} \end{bmatrix}, \quad \begin{bmatrix} \frac{\ell+1}{2} \\ \frac{\ell}{2} \\ \frac{\ell}{2} \\ \frac{\ell-1}{2} \end{bmatrix}, \quad \begin{bmatrix} \frac{\ell}{2} \\ \frac{\ell-1}{2} \\ \frac{\ell-1}{2} \\ \frac{\ell-2}{2} \end{bmatrix}, \tag{5.18}$$

so no matter what $n$ is in (5.15) for the R-charges of $(w_1, w_2, w_3)_P$, there exists a set of R-charges in (5.18) that differ by $\pm\frac{1}{2}$.

## 5.2 Black hole bound states and massive stringy excitations

Depending on whether the constituents are monotone or fortuitous, composite BPS states have different gravitational interpretations. When both $u^i \left|\{(w_1)_{i,\pm\pm}\}\right\rangle$ and $v^j \left|\{(w_2)_{j,\pm\pm}\}\right\rangle$ are monotone, the composite $u^i v^j \left|\{(w_1)_i, (w_2)_j\}\right\rangle$ is also a monotone state that exists in every $N \geq w_1 + w_2$ theory. If either $u^i \left|\{(w_1)_{i,\pm\pm}\}\right\rangle$ or $v^j \left|\{(w_2)_{j,\pm\pm}\}\right\rangle$ is fortuitous in the $N = w_1$ or $N = w_2$ theories, then $u^i v^j \left|\{(w_1)_i, (w_2)_j\}\right\rangle$ is a fortuitous state in the $N = w_1 + w_2$ theory (and a non-BPS state in the $N > w_1 + w_2$ theories).

**Monotone-monotone** These monotone states are dual to supergravitons and smooth horizonless geometries in the bulk by the conjecture of [7].

**Fortuitous-fortuitous: Black hole bound states** It was conjectured [7] that fortuitous states are dual to typical black hole microstates, and hence it is natural to interpret $u^i v^j \left|\{(w_1)_i, (w_2)_j\}\right\rangle$ as a threshold bound state of two black holes, which geometrically arises in the near horizon limit of two-centered black hole solutions [57–59] resulting in e.g. non-trivial fibrations of S$^3$ over AdS.[15]

**Fortuitous-monotone: massive stringy excitations** A state $u^i \left|\{(w_1)_{i,\pm\pm}\}\right\rangle$ representing a fortuitous cohomology class at $N = w_1$ becomes non-BPS for $N > w_1$. In the large $N$ limit ($N \gg w_1$), we expect it to be dual to a massive stringy excitation in vacuum AdS$_3$ × S$^3$ × $T^4$.

---

[15]Two-centered black holes in 4d lift [65, 66] to black rings [67–70] in 5d, and one can consider the near horizon limit of the latter. For the microstate counting of black rings, see [71, 72].

Now, consider the composites of a fortuitous state $u^i \ket{\{(w_1)_{i,\pm\pm}\}}$ with a monotone state $v^j \ket{\{(w_2)_{j,\pm\pm}\}}$, the latter of which is dual to the superstrata geometry [19–26], as discussed in Section 4.2. Hence, in the large $N$ limit ($N = w_1 + w_2 \gg w_1$), it is natural to propose that the bulk dual of the composite state $u^i v^j \ket{\{(w_1)_i, (w_2)_j\}}$ is a massive stringy excitation on the superstrata geometry. Remarkably, the non-BPS massive stringy excitation in the $\text{AdS}_3 \times \text{S}^3 \times T^4$ vacuum becomes BPS on the superstrata background.

Finally, let us emphasize that fortuitous cohomology classes formed by composites of cohomology classes are not typical, as they are multi-cycle in nature. It is expected that single-cycle fortuitous cohomology classes constitute the majority of cohomology classes.

# Acknowledgements

We would like to thank Luis Apolo, Iosif Bena, Nathan Benjamin, Nathan Brady, Yiming Chen, Matthias Gaberdiel, Bin Guo, Samir D. Mathur, Beat Nairz, Zixia Wei, Ashoke Sen, Ergin Sezgin and Xi Yin for helpful discussions. CC is partly supported by the National Key R&D Program of China (NO. 2020YFA0713000). CC thanks the hospitality of Southeast University and Peng Huanwu Center for Fundamental Theory, Hefei, where part of the work was done during the visit. HZ thanks the hospitality of Harvard University and Yau Mathematical Sciences Center, where part of the work was done during the visit. The work of HZ is supported in part by DOE Grant No. DE-SC0010813. This research was supported in part by grant NSF PHY-1748958 to the Kavli Institute for Theoretical Physics (KITP).

# A    Free field realization of $\mathcal{N} = 4$ and covering space calculus

Our convention for the $T^4$ sigma model is that the operator products of the four free complex fields take the form

$$\bar{\alpha}^i(x)\alpha^j(y) \sim \frac{\epsilon^{ij}}{(x-y)^2}, \qquad \bar{\psi}^{\pm}(x)\psi^{\mp}(y) \sim \pm\frac{1}{x-y}. \tag{A.1}$$

The small $\mathcal{N} = 4$ superconformal algebra at $c = 6$ is generated by the currents (with normal ordering suppressed)

$$
\begin{aligned}
T &= \bar{\alpha}^1\alpha^2 - \alpha^1\bar{\alpha}^2 + \frac{1}{2}(-\bar{\psi}^-\partial\psi^+ + \psi^-\partial\bar{\psi}^+ - \psi^+\partial\bar{\psi}^- + \bar{\psi}^+\partial\psi^-), \\
G'^+ &= \bar{\alpha}^2\psi^+ + \alpha^2\bar{\psi}^+, \qquad G^+ = -\bar{\alpha}^1\psi^+ - \alpha^1\bar{\psi}^+, \\
G'^- &= \bar{\alpha}^1\psi^- + \alpha^1\bar{\psi}^-, \qquad G^- = \bar{\alpha}^2\psi^- + \alpha^2\bar{\psi}^-, \\
K^+ &= \bar{\psi}^+\psi^+, \qquad K^- = -\bar{\psi}^-\psi^-, \qquad K^3 = \frac{1}{2}(\bar{\psi}^+\psi^- + \bar{\psi}^-\psi^+).
\end{aligned}
\tag{A.2}
$$

Bosonizing the fermions proves useful for simplifying the calculation of correlation functions in the covering space

$$
\psi^+(z) = e^{i\varphi(z)}, \quad \bar{\psi}^-(z) = -e^{-i\varphi(z)}, \quad \bar{\psi}^+(z) = e^{i\varphi'(z)}, \quad \psi^-(z) = e^{-i\varphi'(z)}
\tag{A.3}
$$

Note that all fermion fields anticommute with each other, whereas the bosonized fields commute. To properly account for this distinction, cocycle factors must be introduced, as discussed in [73, 74].

We provide the corresponding representations of single cycle $\frac{1}{2}$-BPS states in the covering space:

$$
|w_{\pm\pm,\pm\pm}\rangle \leftrightarrow e^{i\frac{w\pm 1}{2}\varphi(z) + i\frac{w\pm 1}{2}\varphi'(z) - i\frac{w\pm 1}{2}\tilde{\varphi}(\bar{z}) - i\frac{w\pm 1}{2}\tilde{\varphi}'(\bar{z})}
\tag{A.4}
$$

The lift of $V(G^-_{-\frac{1}{2}}|\mathrm{BPS}_-\rangle_2)(1)$ to the covering space takes the form:

$$
\bar{\alpha}^1(1)\phi_2^\dagger(1) + \alpha^1(1)\bar{\phi}_2^\dagger(1),
\tag{A.5}
$$

where $\phi_2^\dagger(1)$ and $\bar{\phi}_2^\dagger(1)$ represent the bottom components of the corresponding spin fields. Their bosonizations in the covering space are of form

$$
\phi_2^\dagger(z) = e^{\frac{i}{2}(-\varphi(z) + \varphi'(z) + \tilde{\varphi}(\bar{z}) + \tilde{\varphi}'(\bar{z}))}, \quad \bar{\phi}_2^\dagger(z) = e^{\frac{i}{2}(\varphi(z) - \varphi'(z) + \tilde{\varphi}(\bar{z}) + \tilde{\varphi}'(\bar{z}))}
\tag{A.6}
$$

The following formula is used in the evaluation of fermion correlation functions:

$$
\langle \prod_i e^{i\epsilon_i\varphi(z_i)} \rangle = \prod_{i<j} z_{ij}^{\epsilon_i\epsilon_j}, \qquad \sum_i \epsilon_i = 0
\tag{A.7}
$$

# B  Higher order expansions of partition functions and indices

Expanding the index (2.45), we find the explicit expansion formulae for the $N = 2$ and $N = 3$ modified indices:

$$
\begin{aligned}
I_{\text{NS}}^{\text{Sym}^2(T^4)} ={}& \frac{2}{q^{\frac{1}{2}}} + \left( y + \frac{1}{y} \right) - q^{\frac{1}{2}} \left( 6y^2 + 12 + \frac{6}{y^2} \right) + q \left( y^3 + 39y + \frac{39}{y} + \frac{1}{y^3} \right) \\
& + 2q^{\frac{3}{2}} \left( y^4 - 28y^2 - 96 - \frac{28}{y^2} + \frac{1}{y^4} \right) + q^2 \left( 39y^3 + 513y + \frac{513}{y} + \frac{39}{y^3} \right) \\
& - q^{\frac{5}{2}} \left( 12y^4 + 708y^2 + 2032 + \frac{708}{y^2} + \frac{12}{y^4} \right) \\
& + q^3 \left( y^5 + 513y^3 + 4382y + \frac{4382}{y} + \frac{513}{y^3} + \frac{1}{y^5} \right) + O\left( q^{\frac{7}{2}} \right) ,
\end{aligned}
\tag{B.1}
$$

and

$$
\begin{aligned}
I_{\text{NS}}^{\text{Sym}^3(T^4)} ={}& \frac{3}{q^{\frac{3}{4}}} + q^{\frac{1}{4}} \left( y^2 + 8 + \frac{1}{y^2} \right) - 8q^{\frac{3}{4}} \left( y^3 + 7y + \frac{7}{y} + \frac{1}{y^3} \right) \\
& + q^{\frac{5}{4}} \left( y^4 + 152y^2 + 513 + \frac{152}{y^2} + \frac{1}{y^4} \right) \\
& - 16q^{\frac{7}{4}} \left( 13y^3 + 127y + \frac{127}{y} + \frac{13}{y^3} \right) \\
& + q^{\frac{9}{4}} \left( 3y^6 + 152y^4 + 4382y^2 + 11576 + \frac{4382}{y^2} + \frac{152}{y^4} + \frac{3}{y^6} \right) \\
& - 8q^{\frac{11}{4}} \left( 7y^5 + 702y^3 + 4511y + \frac{4511}{y} + \frac{702}{y^3} + \frac{7}{y^5} \right) + O\left( q^{\frac{13}{4}} \right) .
\end{aligned}
\tag{B.2}
$$

The short characters $\chi_0$, $\chi_1$ of the $c = 12$ contracted large $\mathcal{N} = 4$ superconformal algebra have the expansions

$$
\begin{aligned}
\chi_0 ={}& \frac{1}{q^{\frac{1}{2}}} - 2 \left( y + \frac{1}{y} \right) + q^{\frac{1}{2}} \left( 2y^2 + \frac{2}{y^2} + 9 \right) - 2q \left( y^3 + 9y + \frac{9}{y} + \frac{1}{y^3} \right) \\
& + q^{\frac{3}{2}} \left( y^4 + 22y^2 + 60 + \frac{22}{y^2} + \frac{1}{y^4} \right) - 2q^2 \left( 9y^3 + 55y + \frac{55}{y} + \frac{9}{y^3} \right) \\
& + q^{\frac{5}{2}} \left( 9y^4 + 132y^2 + 305 + \frac{132}{y^2} + \frac{9}{y^4} \right) \\
& - 2q^3 \left( y^5 + 55y^3 + 266y + \frac{266}{y} + \frac{55}{y^3} + \frac{1}{y^5} \right) + O\left( q^{\frac{7}{2}} \right) ,
\end{aligned}
\tag{B.3}
$$

and

$$
\chi_1 = -\left(y + \frac{1}{y}\right) + 2q^{\frac{1}{2}}\left(y^2 + 3 + \frac{1}{y^2}\right) - q\left(y^3 + 15y + \frac{15}{y} + \frac{1}{y^3}\right)
$$
$$
+ q^{\frac{3}{2}}\left(20y^2 + 54 + \frac{20}{y^2}\right) - q^2\left(15y^3 + 113y + \frac{113}{y} + \frac{15}{y^3}\right)
$$
$$
+ q^{\frac{5}{2}}\left(6y^4 + 144y^2 + 338 + \frac{144}{y^2} + \frac{6}{y^4}\right)
$$
$$
- q^3\left(y^5 + 113y^3 + 630y + \frac{630}{y} + \frac{113}{y^3} + \frac{1}{y^5}\right) + O\left(q^{7/2}\right) . \tag{B.4}
$$

The expansion for $\mathcal{S}_1$ in the $N = 2$ BPS partition function (2.51) is

$$
\mathcal{S}_1 = -5\left(y + \frac{1}{y}\right) + 10q^{\frac{1}{2}}\left(y^2 + 3 + \frac{1}{y^2}\right) - 5q\left(y^3 + 15y + \frac{15}{y} + \frac{1}{y^3}\right)
$$
$$
+ 4q^{\frac{3}{2}}\left(25y^2 + 78 + \frac{25}{y^2}\right) - q^2\left(75y^3 + 733y + \frac{733}{y} + \frac{75}{y^3}\right)
$$
$$
+ q^{\frac{5}{2}}\left(30y^4 + 972y^2 + 2642 + \frac{972}{y^2} + \frac{30}{y^4}\right)
$$
$$
- q^3\left(5y^5 + 733y^3 + 5446y + \frac{5446}{y} + \frac{733}{y^3} + \frac{5}{y^5}\right) + O\left(q^{\frac{7}{2}}\right) . \tag{B.5}
$$

The $\mathcal{S}_0^{\mathrm{mon}}$ and $\mathcal{S}_1^{\mathrm{mon}}$ coefficients in the $N = 2$ monotone partition function (4.10) have the expansions:

$$
\mathcal{S}_0^{\mathrm{mon}} = \frac{1}{q^{\frac{1}{2}}} - 2\left(y + \frac{1}{y}\right) + q^{\frac{1}{2}}\left(2y^2 + \frac{2}{y^2} + 9\right) - 2q\left(y^3 + 8y + \frac{8}{y} + \frac{1}{y^3}\right)
$$
$$
+ q^{\frac{3}{2}}\left(y^4 + 14y^2 + 38 + \frac{14}{y^2} + \frac{1}{y^4}\right) - 2q^2\left(3y^3 + 21y + \frac{21}{y} + \frac{3}{y^3}\right)
$$
$$
+ q^{\frac{5}{2}}\left(y^4 + 26y^2 + 70 + \frac{26}{y^2} + \frac{1}{y^4}\right)
$$
$$
- 2q^3\left(5y^3 + 35y + \frac{35}{y} + \frac{5}{y^3}\right) + O\left(q^{\frac{7}{2}}\right) , \tag{B.6}
$$

and

$$
\mathcal{S}_1^{\mathrm{mon}} = -5\left(y + \frac{1}{y}\right) + q^{\frac{1}{2}}\left(10y^2 + 22 + \frac{10}{y^2}\right) - q\left(5y^3 + 38y + \frac{38}{y} + \frac{5}{y^3}\right)
$$
$$
+ q^{\frac{3}{2}}\left(32y^2 + 74 + \frac{32}{y^2}\right) - q^2\left(13y^3 + 91y + \frac{91}{y} + \frac{13}{y^3}\right)
$$
$$
+ 2q^{\frac{5}{2}}\left(y^4 + 32y^2 + 74 + \frac{32}{y^2} + \frac{1}{y^4}\right)
$$
$$
- 21q^3\left(y^3 + 7y + \frac{7}{y} + \frac{1}{y^3}\right) + O\left(q^{\frac{7}{2}}\right).
$$

(B.7)

The $\mathcal{S}_0^{\mathrm{for}}$ and $\mathcal{S}_1^{\mathrm{for}}$ coefficients in the $N = 2$ fortuitous partition function (4.12) have the expansions:

$$
\mathcal{S}_0^{\mathrm{for}} = -2(y^{-1} + y)q + \left(22 + \frac{8}{y^2} + 8y^2\right)q^{3/2}
$$
$$
- 4q^2\left(3y^3 + 17y + \frac{17}{y} + \frac{3}{y^3}\right) + q^{\frac{5}{2}}\left(8y^4 + 106y^2 + 235 + \frac{106}{y^2} + \frac{8}{y^4}\right)
$$
$$
- 2q^3\left(y^5 + 50y^3 + 231y + \frac{231}{y} + \frac{50}{y^3} + \frac{1}{y^5}\right) + +O\left(q^{\frac{7}{2}}\right),
$$

(B.8)

and

$$
\mathcal{S}_1^{\mathrm{for}} = 8q^{\frac{1}{2}} - q\left(37y + \frac{37}{y}\right) + q^{\frac{3}{2}}\left(68y^2 + 238 + \frac{68}{y^2}\right)
$$
$$
- q^2\left(62y^3 + 642y + \frac{642}{y} + \frac{62}{y^3}\right) + q^{5/2}\left(28y^4 + \frac{28}{y^4} + 908y^2 + \frac{908}{y^2} + 2494\right)
$$
$$
- q^3\left(5y^5 + 712y^3 + 5299y + \frac{5299}{y} + \frac{712}{y^3} + \frac{5}{y^5}\right) + O\left(q^{\frac{7}{2}}\right).
$$

(B.9)

The expansion of the $N = 2$ and $3$ monotone indices (4.6) are

$$
I_{N=2}^{\mathrm{mon}} = \frac{2}{q^{\frac{1}{2}}} + \left(y + \frac{1}{y}\right) - q^{\frac{1}{2}}\left(6y^2 + 4 + \frac{6}{y^2}\right) + q\left(y^3 + 6y + \frac{6}{y} + \frac{1}{y^3}\right)
$$
$$
+ 2q^{\frac{3}{2}}\left(y^4 - 2y^2 + 1 - \frac{2}{y^2} + \frac{1}{y^4}\right) + q^2\left(y^3 + 7y + \frac{7}{y} + \frac{1}{y^3}\right)
$$
$$
- q^{\frac{5}{2}}\left(12y^2 + 8 + \frac{12}{y^2}\right) + q^3\left(y^3 + 7y + \frac{7}{y} + \frac{1}{y^3}\right) + O\left(q^{\frac{7}{2}}\right),
$$

(B.10)

and

$$
\begin{aligned}
I_{N=3}^{\mathrm{mon}} =& \frac{3}{q^{\frac{3}{4}}} + q^{\frac{1}{4}}\left( y^2 + 1 + \frac{1}{y^2} \right) - 4q^{\frac{3}{4}}\left( 2y^3 + y + \frac{1}{y} + \frac{2}{y^3} \right) \\
& + q^{\frac{5}{4}}\left( y^4 + 6y^2 + 7 + \frac{6}{y^2} + \frac{1}{y^4} \right) - 4q^{\frac{7}{4}}\left( y^3 + 2y + \frac{2}{y} + \frac{1}{y^3} \right) \\
& + q^{\frac{9}{4}}\left( 3y^6 + y^4 + 7y^2 + 23 + \frac{7}{y^2} + \frac{1}{y^4} + \frac{3}{y^6} \right) \\
& - 4q^{\frac{11}{4}}\left( y^3 + 2y + \frac{2}{y} + \frac{1}{y^3} \right) + O\left( q^{\frac{13}{4}} \right) .
\end{aligned}
\tag{B.11}
$$

The expansion of the $N = 2$ and 3 fortuitous indices (4.8) are

$$
\begin{aligned}
I_{N=2}^{\mathrm{for}} =& -8\sqrt{q} + q\left( 33y + \frac{33}{y} \right) + q^{\frac{3}{2}}\left( -52y^2 - 194 - \frac{52}{y^2} \right) \\
& + q^2\left( 38y^3 + 506y + \frac{506}{y} + \frac{38}{y^3} \right) \\
& + q^{\frac{5}{2}}\left( -12y^4 - \frac{12}{y^4} - 696y^2 - \frac{696}{y^2} - 2024 \right) \\
& + q^3\left( y^5 + \frac{1}{y^5} + 512y^3 + \frac{512}{y^3} + 4375y + \frac{4375}{y} \right) + O\left( q^{\frac{7}{2}} \right) ,
\end{aligned}
\tag{B.12}
$$

and

$$
\begin{aligned}
I_{N=3}^{\mathrm{for}} =& 7\sqrt[4]{q} + q^{\frac{3}{4}}\left( -52y - \frac{52}{y} \right) + q^{\frac{5}{4}}\left( 146y^2 + 506 + \frac{146}{y^2} \right) \\
& + q^{\frac{7}{4}}\left( -204y^3 - 2024y - \frac{2024}{y} - \frac{204}{y^3} \right) \\
& + q^{\frac{9}{4}}\left( 151y^4 + 4375y^2 + 11553 + \frac{4375}{y^2} + \frac{151}{y^4} \right) \\
& + q^{\frac{11}{4}}\left( -56y^5 - 5612y^3 - 36080y - \frac{36080}{y} - \frac{5612}{y^3} - \frac{56}{y^5} \right) + O\left( q^{13/4} \right) .
\end{aligned}
\tag{B.13}
$$

# C  Explicit example of supercharge nilpotency

The nilpotency $Q^2 = 0$ of the supercharge action in first-order conformal perturbation theory is expected, but not manifest from the concrete formulae developed in this paper. As a consistency check, we provide a specific example of the sequential application of $Q^{0,1}$: first acting on $\psi_{-\frac{1}{2}}^+ \bar{\alpha}_{-1}^2 |1_{--}\rangle$, followed by the action of $Q^{0,1}$ on the generated terms. An overall prefactor is factored out from the coefficients discussed in this subsection for better presentation.

The action of supercharge $Q^{0,1}$ on $\bar{\psi}^+_{-\frac{1}{2}}\bar{\alpha}^2_{-1}\left|1_{--}\right\rangle$ generates a linear combination of

$$\alpha^1_{-\frac{1}{2}}\bar{\alpha}^2_{-\frac{1}{2}}\left|2_{--}\right\rangle\,,\ \bar{\alpha}^1_{-\frac{1}{2}}\alpha^2_{-\frac{1}{2}}\left|2_{--}\right\rangle\,,\ \frac{1}{2}\bar{\psi}^+_{-1}\psi^-_0\left|2_{--}\right\rangle\,,\ \frac{1}{2}\psi^+_{-1}\bar{\psi}^-_0\left|2_{--}\right\rangle \tag{C.1}$$

A subsequent action of $Q^{0,1}$ on each of them further source a linear combination of four states

$$\psi^-_{-\frac{1}{6}}\bar{\alpha}^1_{-\frac{1}{3}}\left|3_{--}\right\rangle\,,\ \psi^-_{\frac{1}{6}}\bar{\alpha}^1_{-\frac{2}{3}}\left|3_{--}\right\rangle\,,\ \bar{\psi}^-_{-\frac{1}{6}}\alpha^1_{-\frac{1}{3}}\left|3_{--}\right\rangle\,,\ \bar{\psi}^-_{\frac{1}{6}}\alpha^1_{-\frac{1}{3}}\left|3_{--}\right\rangle \tag{C.2}$$

each with distinct coefficients. The coefficients governing these transitions can be most conveniently expressed in terms of a matrix

$$M = \begin{pmatrix} -1 & -2 & -7 & 10 \\ 7 & -10 & 1 & 2 \\ -7 & -10 & -1 & -2 \\ 1 & 2 & 7 & -10 \end{pmatrix} \tag{C.3}$$

where the component $M_{ij}$ denotes the transition from the $i$ th states in (C.1) to the $j$ th states in (C.2). We see $\sum_i M_{ij} = (0,0,0,0)$ demonstrating $(Q^{0,1})^2$ acts on $\bar{\psi}^+_{-\frac{1}{2}}\bar{\alpha}^2_{-1}\left|1_{--}\right\rangle$ equals to zero.

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
