# Peer review of "Fortuity in the D1-D5 system"

_SciPost Physics_

## Round 2 · Referee Report · Anonymous (Referee 1) · 2025-11-21

Strengths

  1. The lifting of supersymmetric states in the AdS3/CFT2 duality is reformulated in terms of a cohomology problem, and its solution is discussed for finite values of N.
  2. The article is well-written, and it has a nice balance between the original material and review of the literature. This makes the article self-contained.

Weaknesses

The article will benefit from a clear summary of main results and their physical implications.

Report

This interesting article analyzes the fate of supersymmetric states in the two-dimensional orbifold CFT under the deformation that connects the field theory with the supergravity point. It is well-known that some BPS states remain exact under such deformation while others are lifted, and distinguishing these two classes and counting their members is crucial for understanding black hole entropy and microstates which contribute to it. The lifting problem has been extensively studied in the past, and the current article reformulates these known results as a cohomology problem and makes further contribution to the field by studying this problem at a finite value of N, the parameter controlling the field theory. Such study is crucial for connecting the analysis of lifting to recent conjectures about black hole microstates made in reference [7].

The paper performs an impressive technical analysis, and the main results are briefly summarized at the end of the Introduction, but the article would benefit from a more detailed summary either in the discussion section (which is currently missing) or in the introduction. In particular, the current summary ends with the statement "it is natural to propose that the composite state is dual to a BPS stringy excitation on a smooth horizonless geometry corresponding to the cycle-length w2 monotone state.", but does not discuss the physical implications of this result. I recommend this article for publication once all original results are properly formulated and a discussion of their physical implications is added.

Requested changes

All original results should be explicitly formulated in the introduction or discussion sections, and a discussion of their physical implications should be added.

Recommendation

Ask for minor revision

---

## Round 2 · Referee Report · Anonymous (Referee 2) · 2025-12-2

This paper studies 1/4-BPS states in the D1-D5 CFT on $T^4$. They first study the BPS cohomologies for $N = 2$, getting agreement with the index prediction (although they don't relate the result to the notion of fortuity at this stage). Then they propose the notion of and give definitions for fortuitous classes and monotone classes of BPS states, generalizing the similar classification for $\mathcal{N} = 4$ SYM in four dimensions. Monotone classes are further classified into absolute monotone classes and others. They also discuss the possibility of combining two BPS states at smaller $N$ to get a BPS state at larger $N$, relating it potentially to black hole microstates.

This paper contains interesting original contents that are very much relevant for the current research in understanding intrinsic black hole microstates from CFT viewpoints, and deserves publication.

However, in various places, the paper is not exactly easy to read, because the authors do not fully explain some notations and do not say key things that are important for readability. Below I list points to be improved upon before publication:

- At the beginning of section 2.2, the charge assignments of various fields must be clearly and fully explained, or they must explicitly cite a reference where the notation is fully explained. They don't even explain what "$\pm$" on $G'^{\pm}, K^{\pm}$ means.

- They are using two different notations for the automorphism charges, $J$ and $\hat{K}$; they must pick one. On page 8 they introduce $\hat{K}$, but below (2.22) they use $J, \tilde{J}$. In section 5.2, $\tilde{J}$ is used again.

- Footnote 6 is confusing; they should clarify the context.

  It is true that $K_{-1}^{+[i]}$ does not generate new BPS states if one is talking about states on a single strand labeled by $i$, as it is said at the end of the footnote. However, in the first half of the footnote, they talk about SU(2) generators $K_0^3 - \frac{c}{12}$ etc., which sound like the diagonal generator summed over multiple strands. If one is talking about the representation of the *diagonal* generators, then the action of $K_{-1}^{+[i]}$ with a *single* strand *can* produce a new BPS state, belonging to a different SU(2) representation.

  So, they must clarify the context, namely clarify if they are talking about diagonal generators over all strands or generators on one strand.

- In (2.24), they must explain what the signs "$\pm\pm, \pm\pm$" mean. The four signs probably correspond to the number of $\psi^+_{-\frac{1}{2}}, \bar{\psi}^+_{-\frac{1}{2}}, \tilde{\psi}^+_{-\frac{1}{2}}, \tilde{\bar{\psi}}^+_{-\frac{1}{2}}$ excitations, in this order, but they have to say that.

- Below (2.30), I think "$V_p$" should be $V_p^{S_N}$.

- Below (2.30), it says that $V_p^{S_N}$ (assuming the correction in the previous item) forms a representation of $\mathrm{Sym}^N\mathrm{Cliff}$, but it must instead be $(\mathrm{Cliff})^N$, because it is left×right that must be $S_N$ invariant, not the left- and right-moving parts separately. Otherwise, for $N = 2$ for example, one would not get four representations (3.16)–(3.19) but only the symmetric combinations (3.16) and (3.17).

- Above (2.57), it says "a single long multiplet with $j = 0$, $h \geq 0$", but this is misleading. For given $h \geq 0$, there is only one value of $j$ that is allowed ($j = 0$), but of course there are infinitely many long multiplets with different values of $h$.

- Below (3.9), is "$\langle (w+1)_j | Q | w_i \rangle$" must be a typo.

- On pages 24 and 25, starting from the paragraph starting with "The dimensions..." until the end of section 3, it is not obvious what precisely the claim is. Because this is one of the main claims of the paper, they should be clear about it. Was the exact match up to $h = 7/2$ and the states (3.23)–(3.25) obtained by the covering map in section 3.1? Or does the diagram (3.22) lead to the matching without such computations?

- Above (4.3), "the top spaces free BPS sectors" must be a typo.

- In the paragraph containing (4.8), they say that $\pi_{N,N'}^{\mathrm{pre}}$ commutes with the "+" half of the Clifford algebra, but this is incorrect. For example, if one acts with the left-hand side of the first equation of (4.8) on $|1_{--,--}\rangle$ where this $|1_{--,--}\rangle$ is "between" $N$ and $N'$, then one gets zero, while if one acts with the right-hand side on the same state, you get $|1_{--,+-}\rangle$.

- In (4.14), the notation $|i_{--,++}\rangle, |(jk)_{--,--}\rangle, |j_{--,--}\rangle$ is not explained.

- In the second line on page 28, what does the "(1,2) quartet" mean?

- One line above (3.8), I guess that "$V_{(w_1,w_2)P_3}$, $V_{(w_1,w_2)P_4}$, and $V_{(w_1+w_2)}$" need the subscript "top".

- On page 34, in Table 3, the column with $\tilde{j} = \frac{w_1+w_2+w_3+1}{2}$ appears incorrect. The states with $\tilde{J} = -1, 1$ come with degeneracy 1, while the states with $\tilde{J} = 0$ must have degeneracy 3.

There may be more. The authors must carefully check the paper throughout for typos, misspellings, and unclear points.

---

## Editorial Decision

awaiting_resubmission